

# Microphysical sensitivity of coupled springtime Arctic stratocumulus to modelled primary ice over the ice pack, marginal ice, and ocean

Gillian Young[1], Paul J. Connolly[1], Hazel M. Jones[1], and Thomas W. Choularton[1]

[1]Centre for Atmospheric Science, School of Earth and Environmental Sciences, University of Manchester, Manchester, UK.

*Correspondence to:* Gillian Young (gillian.young@manchester.ac.uk)

**Abstract.**

This study uses large eddy simulations to test the sensitivity of single-layer mixed-phase stratocumulus to primary ice number concentrations in the European Arctic. Observations from the Aerosol-Cloud Coupling and Climate Interactions in the Arctic (ACCACIA) campaign are considered for comparison with cloud microphysics modelled using the Large Eddy Model

(LEM, UK Met. Office). We find that cloud structure is very sensitive to ice number concentrations, $N_{ice}$, and small increases can cause persisting mixed-phase clouds to glaciate and break up.

Three key sensitivities are identified with comparison to in situ cloud observations over the sea ice pack, marginal ice zone (MIZ), and ocean. Over sea ice, we find deposition-condensation ice formation rates are overestimated, leading to cloud glaciation. When ice formation is limited to water-saturated conditions, we find microphysics comparable to the aircraft observations

over all surfaces considered. We show that warm supercooled (-13 °C) mixed-phase clouds over the MIZ are simulated to reasonable accuracy when using both the DeMott et al. (2010) and Cooper (1986) parameterisations. Over the ocean, we find a strong sensitivity of Arctic stratus to ice number concentrations. Cooper (1986) performs poorly at the lower ambient temperatures, leading to comparatively higher ice number concentrations ($2.43 \, L^{-1}$ at the cloud top temperature, approximately -20 °C) and cloud glaciation. A small decrease in the predicted $N_{ice}$ ($2.07 \, L^{-1}$ at -20 °C), using the DeMott et al. (2010) pa-

rameterisation, causes mixed-phase conditions to persist for 24 h over the ocean. However, this representation leads to the formation of convective structures which reduce the cloud liquid water through snow precipitation, promoting cloud break up. Decreasing the ice crystal number concentration further ($0.54 \, L^{-1}$, using a relationship derived from ACCACIA observations) allows mixed-phase conditions to be maintained for at least 24 h with more stability in the liquid and ice water paths. Sensitivity to $N_{ice}$ is also evident at low number concentrations, where $0.1 \times N_{ice}$ predicted by the DeMott et al. (2010) parameterisation

results in the formation of rainbands within the model; rainbands which also act to deplete the liquid water in the cloud and promote break up.

## 1 Introduction

The significant uncertainties associated with global climate model (GCM) predictions may be largely attributed to the inadequate treatment of sub-grid scale, such as cloud microphysical, parameterisations (Boucher et al., 2013). These uncertainties





are predicted to enhance discrepancies in temperature forecasts at the polar regions of our planet (ACIA, 2005; Serreze and Barry, 2011; Stocker et al., 2013). The accuracy of these forecasts can be improved by developing the modelled representation of the physical processes involved through comparisons with in situ observations (Curry et al., 1996).

Various observational studies have shown that single-layer mixed-phase stratocumulus (MPS) clouds are common in the

Arctic (e.g. Pinto, 1998; Shupe et al., 2006; Verlinde et al., 2007; Morrison et al., 2012). These clouds have been observed to persist for ~12 h (Shupe et al., 2006) – with some lasting longer than 100 h (Shupe et al., 2011) – whilst maintaining cloud top temperatures as low as -30 °C (Verlinde et al., 2007). Single-layer Arctic MPS typically form at low altitudes and maintain a liquid layer at cloud top which facilitates ice formation and precipitation below (Rangno and Hobbs, 2001; Shupe et al., 2006; Verlinde et al., 2007; McFarquhar et al., 2011; Jackson et al., 2012; Morrison et al., 2012, amongst others). The

Wegener-Bergeron-Findeisen (WBF) mechanism strongly influences MPS and initiates a continually-changing microphysical structure. Moderate vertical motions maintain these clouds, where mixing ensures that the proximity between ice crystals and cloud droplets is variable whilst sustaining supersaturated conditions (Korolev and Isaac, 2003).

Models do not reproduce the microphysical structure and radiative interactions of these persistent Arctic mixed-phase clouds well (e.g. Tjernström et al., 2008; Klein et al., 2009; Morrison et al., 2009; Morrison et al., 2012; de Boer et al., 2014). Detailed

cloud resolving model (CRM) simulations have previously shown that commonly-used mid-latitude parameterisations, such as Cooper (1986) or Meyers et al. (1992), overestimate the cloud ice number concentration, $N_{ice}$, in Arctic MPS, causing the rapid depletion of liquid and cloud glaciation (Harrington et al., 1999; Prenni et al., 2007). Modelled MPS are particularly sensitive to $N_{ice}$, with small decreases in simulated ice number causing significant increases in modelled liquid water path (Harrington and Olsson, 2001).

Four ice nucleation modes are commonly represented in models, and three (immersion-, contact-, and condensation-freezing) require the presence of cloud droplets for initiation (Pruppacher and Klett, 1997). Immersion-freezing occurs when a cloud droplet is nucleated by an aerosol particle of mixed composition; mixing which likely incorporates both soluble and insoluble fractions. Solubility is a crucial property of an efficient cloud condensation nuclei (CCN) whilst efficient ice nucleating particles (INPs) are insoluble (Pruppacher and Klett, 1997; Murray et al., 2012). The inclusion of an insoluble fraction would allow a

CCN to obtain some ability as an INP (de Boer et al., 2010). In the atmosphere, soluble coatings on previously CCN-inactive particles, like desert dusts, promote ice nucleation via this pathway (Bigg and Leck, 2001). For example, organic coatings can suppress the ability of an INP to nucleate via the deposition mode (Möhler et al., 2008; Primm et al., 2016). Deposition-freezing results from the direct deposition of water vapour onto an INP, and is often linked with condensation-freezing due to difficulties in distinguishing these mechanisms in measurements. Deposition-freezing may occur in both water- and ice-

supersaturated conditions. The frequency of MPS in the Arctic suggests that ice formation in these clouds is tied to the liquid phase, as preferential nucleation via deposition-freezing may, in theory, result in a higher proportion of fully glaciated clouds than are observed (de Boer et al., 2011; Vihma et al., 2014). Consequently, recent studies (e.g. de Boer et al., 2011) suggest that liquid-dependent modes of nucleation are dominant in Arctic MPS at sub-zero temperatures greater than -25 °C.

Liquid-dependent freezing may be inferred by observations in the Arctic. Previous studies have found correlations between

the number concentrations of ice crystals and large (>23 μm) cloud drops (and drizzle drops, Hobbs and Rangno, 1998; Rangno



and Hobbs, 2001). These large liquid particles have an increased likelihood of containing a partially-insoluble nucleus, or colliding with one, due to aerosol scavenging; therefore, they may nucleate via immersion- or contact-freezing respectively. Arctic aerosol particles are often well-mixed due to long-range transport (Young et al., 2016b); therefore, they can provide a efficient platform for immersion-freezing (Bigg and Leck, 2001; de Boer et al., 2010). Similarly, mixed particles can promote ice nu-
cleation through collisions with cloud droplets; however, contact-freezing nuclei are generally thought to be predominantly insoluble and ice-active, with little CCN ability (Young, 1974).

Investigating the sensitivity of MPS to ice crystal number concentrations will help to improve our understanding of the microphysical limitations of these clouds. Here, we test if primary ice formation under water-saturation conditions improves the modelled microphysical structure with comparison to deposition-condensation freezing, with the hypothesis that ice number
concentrations will be suppressed under this restriction. Modelling studies which utilise immersion-freezing have successfully simulated the persistence of Arctic stratocumulus clouds, producing sustained liquid water in the presence of ice crystals for up to 12 h (de Boer et al., 2010).

Here, we use in situ cloud observations of Arctic MPS, from the Aerosol-Cloud Coupling and Climate Interactions in the Arctic (ACCACIA) campaign of 2013, as a guide to infer the microphysical sensitivity of modelled clouds to both ice number
and surface conditions. We use the Large Eddy Model (LEM, UK Met Office, Gray et al., 2001) to simulate cloud microphysics observed over the sea ice, marginal ice zone (MIZ), and ocean. The UK's BAe-146-301 Atmospheric Research Aircraft was used during the springtime (Mar-Apr) campaign, collecting high-resolution in situ observations of the cloud microphysics encountered (Lloyd et al., 2015; Young et al., 2016a). Several dropsondes were launched from the aircraft during these cases to provide vertical profiles of the boundary layer (BL) structure. By combining dropsonde and in situ measurements, the sensi-
tivity of modelled cloud microphysics to changes in predicted ice number concentrations is tested to infer the microphysical limitations of persistent springtime MPS in the European Arctic.

## 2 Methodology

### 2.1 Aircraft Instrumentation

Measurements from instruments on-board the Facility for Airborne Atmospheric Measurements' (FAAM) BAe-146 aircraft
during three chosen case studies are presented to test the ability of the LEM to reproduce the Arctic mixed-phase clouds observed. Specifically, data from two wing-mounted instruments – the 2-Dimensional Stereo Particle imaging probe (2DS, Lawson et al., 2006) and Cloud Droplet Probe (CDP-100 Version 2, Droplet Measurement Technologies (DMT), Lance et al., 2010) – are used to investigate mixed-phase clouds, as these probes can measure the sizes and number concentrations of ice crystals (80-1280 μm) and cloud droplets (3-50 μm) respectively. Details on the functioning of these probes, data analysis, and
subsequent particle phase discrimination have been discussed previously by Crosier et al. (2011, 2014) and Taylor et al. (2016). The use of these instruments during ACCACIA is discussed by Lloyd et al. (2015) and Young et al. (2016a).

Aerosol particle data are used for the evaluation of the DeMott et al. (2010) ice nucleation parameterisation. Data from the Passive-Cavity Aerosol Spectrometer Probe (PCASP 100-X, Droplet Measurement Technologies, Rosenberg et al., 2012) are



used to size and count aerosol particles from sizes 0.1 µm to 3 µm. Additionally, dropsondes released during each case are used to provide representative vertical profiles of potential temperature, water vapour mixing ratio, and wind fields to initialise the model.

## 2.2 Large Eddy Model (LEM)

The LEM allows cloud microphysics to be studied in isolation from large scale meteorological features. Cloud microphysical interactions, wind velocities, and turbulent motions within the boundary layer are simulated to allow a detailed investigation of cloud formation and evolution over the domain (Boucher et al., 2013). Here, we consider three case studies of observations over the sea ice, marginal ice zone, and ocean; cases 1, 2, and 3 respectively.

A 16 km×16 km domain was used, centred on the respective dropsonde release points in each case, with a spatial resolution
of 120 m and a model height of 3 km applied. A vertical resolution of 20 m was imposed from the surface to the altitude of the boundary layer temperature inversion (1500 m), above which it was reduced to 50 m. The LEM was run for 24 hours to simulate the respective observations. The first 3 hours of each simulation was not considered due to model spin-up. For all cases, cyclic lateral boundary conditions were imposed. A sponge layer was applied to the top 500 m of the domain, allowing the fields to revert back to their initial conditions in this region. Long- and shortwave radiation was modelled using the Edwards
and Slingo (1996) scheme and was called every 150 seconds within the model. Dropsonde profiles of potential temperature, wind speed, and water vapour mixing ratio were used to initialise the model. An adiabatic liquid water profile was assumed up to the first temperature inversion (approximately 600 m, 350 m, and 1150 m for cases 1, 2, and 3 respectively).

Over the ocean and marginal ice zone (cases 2 and 3), surface fluxes were calculated by the model, which assumes a water-saturated, ocean surface. Small sensible heat fluxes ($1\,\mathrm{W\,m^{-2}}$) were imposed to simulate the sea ice surface (case 1), as studies
have measured such values adjacent to the ice pack (e.g. Sotiropoulou et al., 2014). A sub-Arctic McClatchy profile was imposed in all simulations to ensure the initialised vertical profiles of tropospheric temperature, pressure, water vapour, and ozone were representative of the environment modelled.

No large-scale subsidence was imposed in these simulations to allow the microphysical effect of ice number and surface fluxes to be studied in isolation. Imposed subsidence would affect the microphysical structure of the modelled clouds, and the
effect of including large-scale subsidence is discussed in Sect. 5.4.1.

### 2.2.1 Primary Ice Nucleation

The double-moment microphysics scheme by Morrison et al. (2005) is used within the LEM to test the sensitivity of the simulated mixed-phase Arctic clouds to ice number concentration. This scheme represents single-moment liquid, with a prescribed droplet number, and double-moment ice, snow, graupel, and rain. Quoted $N_{ice}$ in this article represents the summed
contributions of the ice crystal, graupel, and snow number concentrations simulated. 2DS measurements are not segregated into such categories; therefore, bulk, "total ice" number concentrations are compared. A mean droplet number of $100\,\mathrm{cm^{-3}}$, approximated from the aircraft observations, is applied in all simulations. The sensitivity of the ice phase to this number is not considered here.





Deposition-condensation, immersion-, and contact-freezing are all represented within the Morrison microphysics scheme. The form of the deposition-condensation freezing parameterisation is varied in this study to test the cloud microphysical response. Immersion-freezing is included as the Bigg parameterisation (Bigg 1953 - hereafter, B53) and contact-freezing is represented by the Meyers parameterisation (Meyers et al. 1992 - hereafter, M92). The influence of these parameterisations on simulated ice number concentrations is detailed in the Supplement. To investigate the sensitivity of the modelled microphysics to predictable primary ice number concentrations, B53 immersion- and M92 contact-freezing were switched off within the microphysics scheme, and the sole contribution to $N_{ice}$ from one implemented parameterisation was considered.

Three distinct ice nucleation parameterisations were imposed in this study (Fig. 1). Firstly, the deposition-condensation ice nucleation parameterisation proposed by Cooper 1986 (hereafter, C86) was tested against the ACCACIA observations. This relationship is commonly used within the Morrison microphysics scheme in the Weather Research and Forecasting (WRF) model, amongst others. In Eq. 1, $N_{ice}$ represents the number concentration of pristine ice crystals, and $T_0 - T_K$ defines the sub-zero temperature. This parameterisation is used to simulate ice number concentrations below 265 K only.

$$N_{ice}[m^{-3}, T_K] = 5 \cdot \exp\left(0.304\left[T_0 - T_K\right]\right) \tag{1}$$

Secondly, an approximation of the DeMott et al. 2010 (hereafter, D10) parameterisation was applied. This study derived a detailed relationship between INP number, temperature, and aerosol number concentration based on an amalgamation of different INP field data. D10 was imposed at temperatures below 264 K and at water-saturation (in accordance with DeMott et al., 2010).

$$N_{INP}[m^{-3}, T_K] = 0.0594\left(273.16 - T_K\right)^{3.33}\left(n_{aer,0.5}\right)^{0.0264(273.16 - T_K) + 0.0033} \tag{2}$$

Equation 2 predicts the number concentration of INPs active at the given temperature, $T_K$. As input, it requires $n_{aer,0.5}$: the number concentration of aerosol particles with diameter, $D_P$, greater than 0.5 μm. These aerosol data were averaged using PCASP measurements in the close vicinity to the observed cloud, producing input concentrations of 1.13 cm$^{-3}$, 1.77 cm$^{-3}$, and 2.20 cm$^{-3}$ over the sea ice, MIZ, and ocean respectively. Below-cloud data were solely used over the ocean, whereas above-cloud measurements were included in the sea ice and MIZ calculations as the observed clouds had sub-adiabatic liquid water profiles, making entrainment processes likely.

Additionally, a curve was fitted to the observed ice crystal number concentrations during ACCACIA and used within the model. Data from ACCACIA flights B761, B762, B764, B765, and B768 are included in the derivation of this curve. Microphysical data from B762, and B761/B768, have been previously detailed by Young et al. (2016a) and Lloyd et al. (2015)





**Table 1.** Predicted $N_{ice}$ [L$^{-1}$] using each parameterisation considered in this study at the observed cloud top temperatures in each case.

| Case Number | Temperature[a] [K (°C)] | C86 | D10[b] | ACC | D10*×0.1 | D10*×10 |
|---|---|---|---|---|---|---|
| 1 | 253.4 (-19.8) | 2.03 | 1.31 | 0.51 | 0.13 | 13.1 |
| 2 | 260.5 (-12.7) | 0.23 | 0.34 | 0.17 | 0.03 | 3.37 |
| 3 | 252.8 (-20.4) | 2.43 | 2.07 | 0.54 | 0.21 | 20.7 |

[a] Cloud top temperature (CTT)

[b] $N_{INP,L^{-1}}$

respectively. Young et al. (2016b) illustrate the corresponding flight tracks of each of these cases. Bulk number concentrations from these flights were plotted against temperature and the following relationship was derived from these data:

$$N_{ice}[m^{-3}, T_K] = \frac{0.068\left(273.5 - T_K\right)^{3.3}}{\exp\left(0.05\left(273.16 - T_K\right)\right)} \tag{3}$$

This curve is valid below 265 K. Temperatures greater than this were subject to minor secondary ice production (see Young
et al., 2016a); therefore, the primary ice component could not be cleanly extracted from these data. These observed ice data
spanned 252 K to 265 K. This curve somewhat mirrors the shape of D10 (Fig. 1); however, it is weighted by an exponential
term to provide better agreement with the observations at low temperatures. In this article, this curve will be abbreviated to
ACC.

INPs are not depleted in this study; however, ice crystal number concentrations are prognostic within the Morrison et al.
(2005) microphysics scheme. Aerosol particles are not strictly represented in the LEM and the microphysical representation is
bulk, not binned. These simulations are only representative of a system with a replenishing source of INPs, and are therefore
idealistic representations of the modelled clouds.

The primary objective of this study is to identify the sensitivity of cloud stability to ice crystal number concentration. DeMott
et al. (2010) suggest that INP number concentrations need to be predicted to within a factor of 10 to avoid an unrealistic
treatment of mixed-phase cloud microphysics. Therefore, D10×10 and D10×0.1 were considered – in addition to C86, D10,
and ACC – to additionally test sensitivity of simulated mixed-phase cloud microphysics to large changes in ice crystal number
concentration. Figure 1 illustrates the performance of each parameterisation considered: the C86 and ACC cases, dependent
only on temperature, are valid across the three observational studies chosen, whilst the D10 parameterisation – and variations
thereof – is variable between cases given its dependence on observed aerosol particle number concentrations.




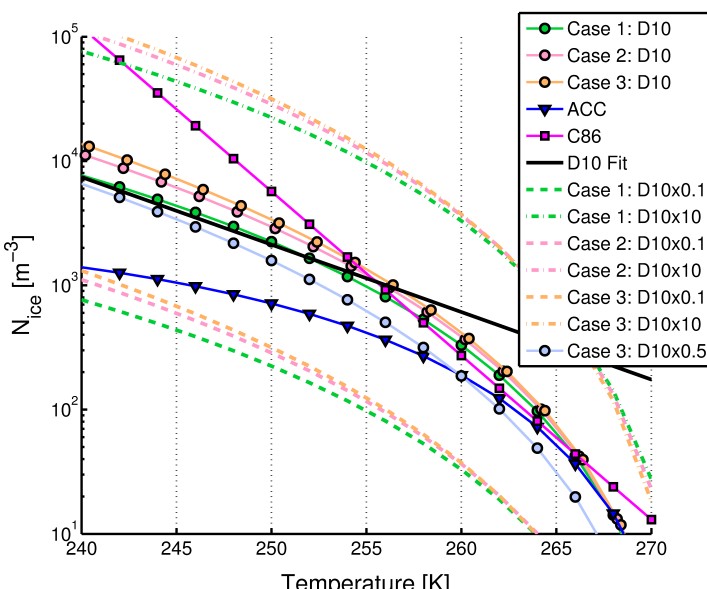

**Figure 1.** Evaluation of the five parameterisations used (C86, D10, ACC, D10×0.1, and D10×10) in the three cases considered with respect to temperature. **Case 1:** sea ice, **case 2:** marginal ice zone (MIZ), and **case 3:** ocean. The black line depicts the temperature-dependent fit of DeMott et al. (2010) for reference. The C86 parameterisation and ACC are valid for all cases, whereas the different aerosol particle loadings are accounted for with the D10 parameterisation. D10×0.5 is implemented in the ocean case in Sect. 5.4.2.

## 3   Aircraft observations

In situ observations of cloud microphysics over the sea ice and ocean during ACCACIA flight B762 (23 Mar 2013), and over the marginal ice zone (MIZ) during flight B764 (29 Mar 2013), are considered for model comparison. Microphysical observations from flight B762 have been detailed previously by Young et al. (2016a). The corresponding flight tracks are illustrated in

5   Fig. 2. These case studies were chosen due to the availability of dropsondes for model initialisation and temporally-close in situ aircraft observations. Dropsondes from B762 distinctly sampled either the sea ice or ocean (as shown in Fig. 2**a**). The ocean dropsonde was far from the sea ice edge (∼140 km). The B764 dropsonde (Fig. 2**b**) was dropped over the MIZ. As in Young et al. (2016a), the MIZ was defined as sea ice fractions >10 % and <90 % based on NSIDC data (National Snow and Ice Data Centre, Fig. 2). These three cases were conducted over similar longitudes (∼27 °E) and approximately the same latitude

10   range (∼75-77 °N).

Figure 3 shows the potential temperature, vapour, and wind speed profiles measured by each dropsonde used to initialise the LEM. In all cases, the net wind direction was north-easterly, bringing cold air from over the sea ice pack to the over the





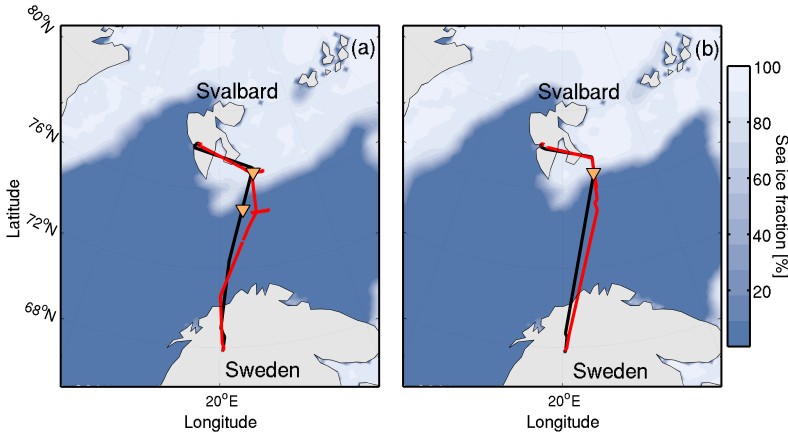

**Figure 2.** Flight track of **(a)** B762 and **(b)** B764, with section 1 (black) and section 2 (red) indicated. Dropsondes were released during section 1, whilst in situ observations were made during section 2. Dropsonde release locations are marked (orange triangles). Case 1 (sea ice, north) and case 3 (ocean, south) are from flight B762, whilst case 2 (MIZ) is from flight B764. Sea ice fraction is shown in shading.

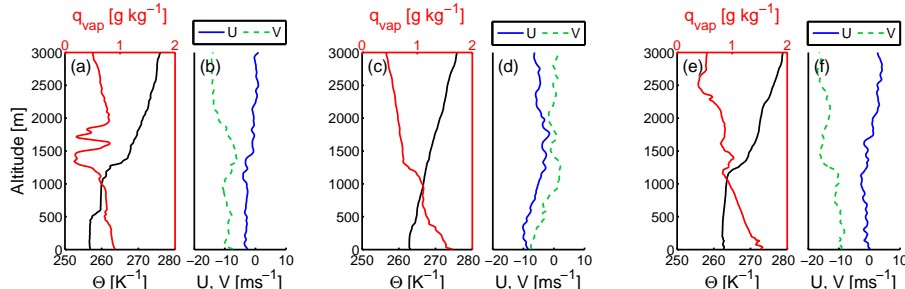

**Figure 3.** Potential temperature, vapour mixing ratio, and wind speed profiles measured by the three dropsondes used to initialise the LEM in this study. **(a-b):** dropsonde 1 released over the sea ice during flight B762. **(c-d):** dropsonde 2 released over the MIZ during flight B764. **(e-f):** dropsonde 3 released over the ocean during flight B762.

comparatively warm ocean. The potential temperature profile for the sea ice case (case 1) displays a double inversion; the first at ∼600 m and the second at ∼1100 m. The latter inversion is at approximately the same altitude as that measured in the ocean case (case 3). The MIZ case shows a subtle inversion at approximately 500 m; however, it is not as prominent as the other two cases.




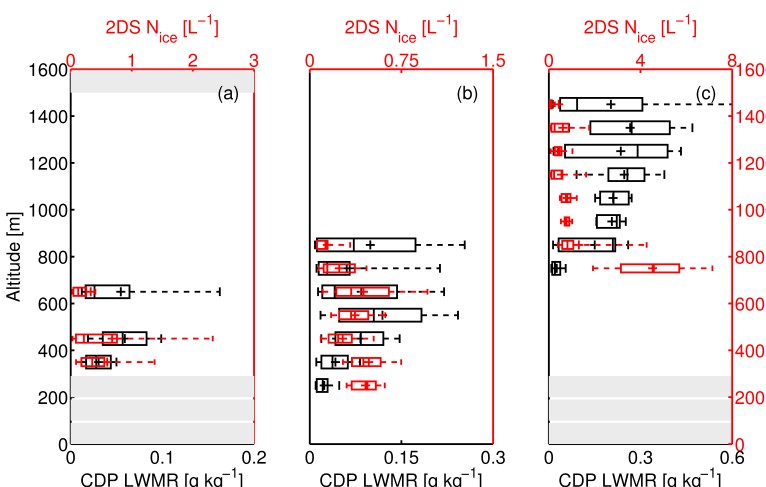

**Figure 4.** Observations of 2DS ice number concentration (red) and CDP liquid water mixing ratio (LWMR, black). **(a):** Sea ice, case 1. **(b):** MIZ, case 2. **(c):** Ocean, case 3. Only observations from mixed-phase cloud are included, with a derived CDP liquid water content threshold of $\geq 0.01 \, \mathrm{g \, m^{-3}}$ distinguishing in-cloud measurements. Box edges: $25^{\mathrm{th}}$ and $75^{\mathrm{th}}$ percentiles, Median: | , and Mean: +. Altitudes not sampled by the aircraft are indicated with grey boxes.

**Table 2.** Summary of cloud observations for each of the three cases considered. Values quoted are averaged quantities, with $1\sigma$ in brackets.

| Case Number | Flight | Date [2013] | Surface Conditions | Cloud Extent [m] | LWMR[a] [g kg$^{-1}$] | $N_{\mathrm{ice}}^{[b]}$ [L$^{-1}$] |
|---|---|---|---|---|---|---|
| 1 | B762 | 23 Mar | Sea ice | 300-700 | 0.05 (0.04) | 0.47 (0.86) |
| 2 | B764 | 29 Mar | MIZ/Ocean | 200-900 | 0.09 (0.07) | 0.35 (0.20) |
| 3 | B762 | 23 Mar | Ocean | 700-1500 | 0.24 (0.13) | 0.55 (0.95) |

[a] Liquid water mixing ratio

[b] Ice crystal number concentration

In situ measurements for all cases show a distinct, mixed-phase cloud from approximately 300 m to 700 m (case 1), 200 m to 900 m (case 2) and 700 m to 1500 m (case 3, Fig. 4). These measurements are summarised in Table 2. Liquid water mass mixing ratios (LWMR), derived from CDP measurements, provide a direct comparison with the LEM: the liquid measurements in the sea ice case are low, of the order of $\sim$0.05 g kg$^{-1}$, whereas the MIZ and ocean cases have larger mixing ratios

5    ($\sim$0.1-0.3 g kg$^{-1}$). 2DS ice number concentrations are consistently low within the cloud layer in all cases, on the order of approximately 0.2-1.5 L$^{-1}$. High ice number concentrations at cloud base in case 3 are thought to be minor contributions of



secondary ice due to crystal fragmentation (Young et al., 2016a). Cloud top temperatures (CTTs) were approximately -20°C, -13°C and -20°C respectively (Table 1). Such temperatures are too cold for efficient secondary ice production and too warm for homogeneous ice nucleation (Hallett and Mossop, 1974; Pruppacher and Klett, 1997). For this study, modelled microphysics below 1500 m is focused upon as this is directly comparable with these aircraft observations.

## 4 Results

### 4.1 Control simulations

Within the Morrison et al. (2005) bulk microphysics scheme, the C86 ice nucleation parameterisation is used to simulate the heterogeneous nucleation of ice. Onset conditions commonly used in the WRF model (T < -8°C and $S_w$ > 0.999, or $S_i$ > 1.08) were applied as a control simulation for each case. Figure 5 shows the ice number concentrations, $N_{ice}$, and liquid water mixing ratios, $Q_{liq}$, modelled over the sea ice (case 1), MIZ (case 2), and ocean (case 3). In case 1, no liquid water is modelled (Fig. 5a). Ice number concentrations of $\sim$3 L$^{-1}$ are simulated at an altitude of approximately 1000 m for the first 10 h of the run, peaking at 3.4 L$^{-1}$. This ice then dissipates, after which $N_{ice} \sim$1 L$^{-1}$ is maintained at 500 m for the remainder of the simulation. This sustained number concentration is within the range observed (0.47 ± 0.86 L$^{-1}$, Table 2); however, mixed-phase conditions are not modelled. In contrast, co-existing regions of liquid and ice are simulated in cases 2 and 3. Modelled $N_{ice}$ over the MIZ ($\sim$1.0 L$^{-1}$, Fig. 5b) is in reasonable agreement with the mean observed (0.35 ± 0.20 L$^{-1}$, Table 2). Persistent mixed-phase conditions are simulated in case 2 for approximately 16 h. Such conditions are also attained in case 3 (Fig. 5c), with modelled ice number concentrations much greater than observed; modelled $N_{ice}$ peaks at 3.7 L$^{-1}$, compared with 0.55 ± 0.95 L$^{-1}$ observed. This case glaciates after approximately 15 h. Therefore, under the conditions commonly used in the WRF model, C86 overpredicts $N_{ice}$ and unsuccessfully reproduces the observed mixed-phase conditions over all three surfaces considered. To force the formation of persistent liquid in all cases, we restrict the formation of primary ice to water-saturated conditions in our subsequent model runs.

### 4.2 Ice nucleation at water-saturation

#### 4.2.1 Case 1: Sea ice

Figure 6(**a, d, g**) shows modelled $N_{ice}$ and liquid water mixing ratio, $Q_{liq}$, using the three main parameterisations – C86, D10, and ACC – over the sea ice. Vertical (Z-Y) slices of $N_{ice}$, $Q_{liq}$, and W at 21 h are included in the Supplement (Fig. S2). A mixed-phase cloud is simulated at $\sim$500 m after 17 h, with a liquid layer at cloud top with ice formation and precipitation below. Peak $Q_{liq}$ varies from C86 at the smallest (0.09 g kg$^{-1}$), through D10 (0.1 g kg$^{-1}$), to ACC at the largest (0.14 g kg$^{-1}$, Table 3). $N_{ice}$ and $Q_{liq}$ increase with time as the cloud evolves. Modelled $N_{ice}$ is of the same order of magnitude using each parameterisation, with maximum values of 2.32 L$^{-1}$, 1.29 L$^{-1}$, and 0.47 L$^{-1}$ attained by C86, D10, and ACC respectively.

Figure 7 shows a comparison between measured and modelled (total) $N_{ice}$, $N_{ice>100\mu m}$, and $Q_{liq}$ for each case when using these three parameterisations. Comparisons including D10×10 and D10×0.1 are included in the Supplement (Fig. S5). Mean



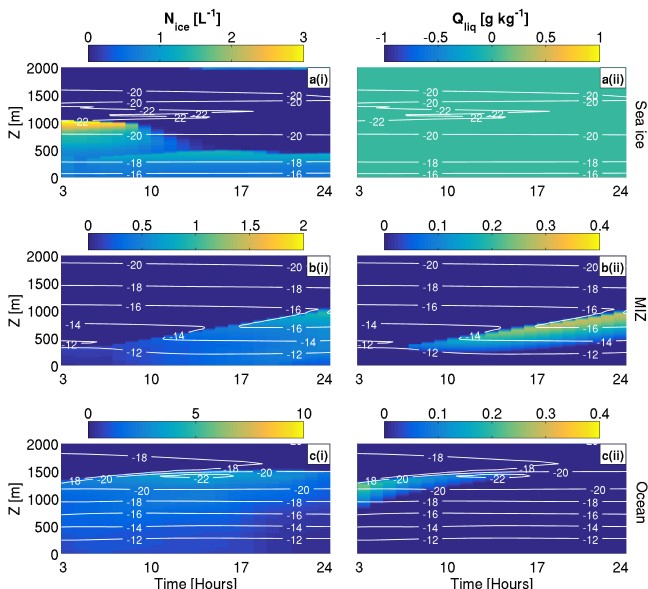

**Figure 5.** Simulated ice number concentrations ($N_{ice}$, **(i)**) and liquid water mixing ratios ($Q_{liq}$, **(ii)**) using the C86 parameterisation under default WRF conditions ($T < -8°C$, $S_w > 0.999$, or $S_i > 1.08$). **a:** Sea ice (case 1), **b:** MIZ (case 2), **c:** Ocean (case 3). Run length 24 hours. Temperature (°C) contours are overlaid in white. Note changing colour bar for each subfigure.

**Table 3.** Maximum modelled values during each case for each parameterisation implemented at water-saturation.

| Case | Parameter | C86 | D10 | ACC | D10×10 | D10×0.1 |
|---|---|---|---|---|---|---|
| Sea ice (case 1) | $N_{ice}$ [L$^{-1}$] | 2.32 | 1.29 | 0.47 | 2.89 | 0.13 |
| | $Q_{liq}$ [g kg$^{-1}$] | 0.09 | 0.10 | 0.14 | 0 | 0.16 |
| MIZ (case 2) | $N_{ice}$ [L$^{-1}$] | 1.09 | 1.03 | 0.36 | 6.57 | 0.11 |
| | $Q_{liq}$ [g kg$^{-1}$] | 0.29 | 0.28 | 0.34 | 0.12 | 0.39 |
| Ocean (case 3) | $N_{ice}$ [L$^{-1}$] | 3.83 | 3.01 | 0.71 | 15.5 | 0.37 |
| | $Q_{liq}$ [g kg$^{-1}$] | 0.32 | 0.32 | 0.36 | 0.10 | 0.38 |

parameters modelled at 21 h during case 1 are shown in Fig.7(**a, d, g**). The method for choosing these time steps is detailed in the Supplement (Figs. S6). C86 produces the greatest ice number concentration, with D10 producing the second greatest and ACC producing the least (Fig. 7**a**). ACC provides the best agreement with the mean observed $N_{ice}$, simulating approximately $0.4\,L^{-1}$.





Software for processing the 2DS data cannot distinguish between liquid and ice at small sizes ($<80\,\mu m$); therefore, the number concentration of small ice crystals is not a reliable measure with this instrument. For this reason, the observed number concentration of ice crystals greater than $100\,\mu m$ are also compared with those modelled in this size range. Figure 7**d** shows this comparison for case 1. Again, ACC performs well, with approximately $0.2\,L^{-1}$ simulated. D10 and C86 produce a larger

$N_{ice>100\mu m}$ than observed. Figure 7**g** shows the comparison of observed LWMR and modelled $Q_{liq}$. Modelled variability is not as clear as with the $N_{ice}$ data, and most of the variability occurs at the same altitude ($500\,m$). In contrast to $N_{ice}$ and $N_{ice>100\mu m}$, ACC produces the greatest $Q_{liq}$, while C86 and D10 underestimate with respect to the observations.

Modelled $N_{ice}$ and $Q_{liq}$ for D10$\times$10 and D10$\times$0.1 are shown in Fig. 8(**a, g**), and D10 is again included (Fig. 8**d**) for comparison. D10$\times$0.1 produces $N_{ice}$ values which are approximately a factor of 2 too low. As a consequence of these lower

ice number concentrations, $Q_{liq}$ is enhanced, with maximum of $0.16\,g\,kg^{-1}$ modelled. In contrast, no liquid water is simulated when using D10$\times$10, with peak ice number concentrations of $2.89\,L^{-1}$ produced at approximately $400\,m$.

Liquid and ice water paths (LWP and IWP, respectively) are shown in Fig. 9(**a, d**). Both increase with model time when using each of the parameterisations. D10$\times$0.1 produces the highest LWP and lowest IWP. D10$\times$10 produces no liquid – giving a LWP of zero – and the simulated IWP increases initially (between approximately $17\,h$ and $20\,h$), but subsequently decreases.

The D10 and C86 parameterisations produce similar trends in the LWP and IWP traces, resulting in approximately $15$-$20\,g\,m^{-2}$ and $2$-$3\,g\,m^{-2}$ respectively by $24\,h$.

### 4.2.2    Case 2: Marginal ice zone

Figure 6(**b, e, h**) shows that there is little variation between the simulations over the MIZ (case 2). C86 and D10 produce comparable peak $N_{ice}$ and $Q_{liq}$ values (Table 3). These parameterisations also produce a similar LWP ($\sim 100\,g\,m^{-2}$) and IWP

($\sim 7\,g\,m^{-2}$) by the end of each simulation (Fig. 9**b, e**). More liquid and less ice is simulated with ACC (Fig. 6**h**, Table 3). D10$\times$0.1 produces the lowest $N_{ice}$ overall ($0.11\,L^{-1}$, Fig. 8**h**). This allows $Q_{liq}$ to be greater than in the other simulations ($0.39\,g\,kg^{-1}$, Table 3) and the simulated LWP and IWP increase steadily with time (Fig. 9**b, e**). D10$\times$0.1 produces the lowest IWP, whilst D10$\times$10 produces the greatest. $N_{ice}$ of up to $6.6\,L^{-1}$ are simulated using D10$\times$10, with a suppressed $Q_{liq}$ (Fig. 8**b**).

At $17\,h$, $N_{ice}$ modelled using ACC are lower ($0.2\,L^{-1}$) in comparison to the mean observed at each altitude bin (Fig. 7**b**).

However, ACC overpredicts $N_{ice>100\mu m}$ compared to observations ($0.13\,L^{-1}$ versus $0.03\,L^{-1}$, Fig. 7**e**). D10 produces the greatest $N_{ice}$ in this case, yet C86 produces a similar concentration. Again, D10 and C86 both overpredict $N_{ice>100\mu m}$ and variability in $Q_{liq}$ is limited to the same altitude ($700\,m$, Fig. 7**h**).



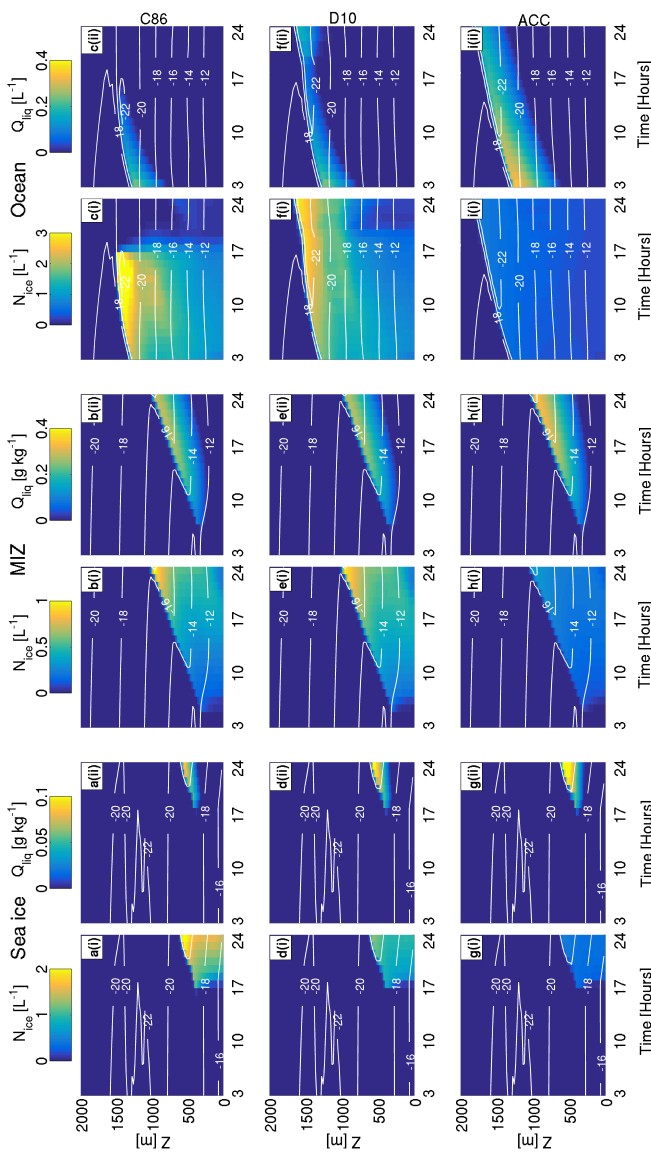

**Figure 6.** Simulated ice number concentrations ($N_{ice}$, **(i)**) and liquid water mixing ratios ($Q_{liq}$, **(ii)**) using the **(a-c)** C86, **(d-f)** D10, and **(g-i)** ACC parameterisations. All are restricted to water-saturation. **(a, d, g):** Sea ice (case 1), **(b, e, h):** MIZ (case 2), **(c, f, i):** Ocean (case 3). Run length 24 hours. Temperature (°C) contours are overlaid in white. Colour bar at the top of each column corresponds to data in that column only.

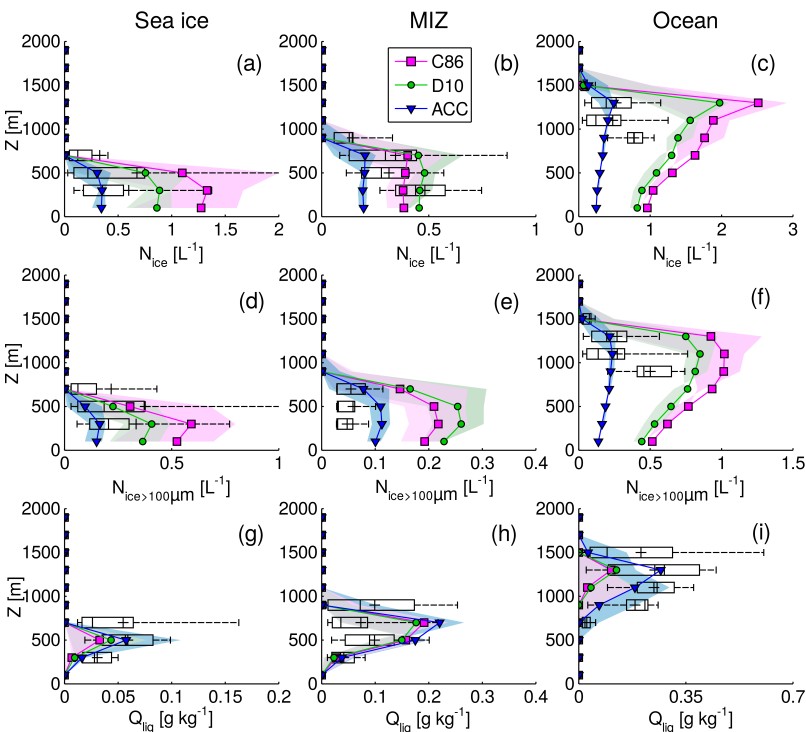

**Figure 7.** Observed $N_{ice}$, $N_{ice>100\mu m}$, and $Q_{liq}$ for the sea ice (**column 1**), MIZ (**column 2**), and ocean (**column 3**) cases. Observations are shown as black boxes. Box edges represent the 25th and 75th percentiles, and the median and mean values are denoted by | and + respectively. Mean modelled values using the C86 (magenta), D10 (green), and ACC (blue) parameterisations are overlaid. Model time steps of 21 h, 17 h, and 7 h are used for the sea ice, MIZ, and ocean cases respectively, as these time steps offer the best comparison with the observations. Shading (in pink, green, or blue for C86, D10 and ACC respectively) indicates variability in the model parameters from $\pm 3$ h in cases 1 and 2, and $\pm 4$ h in case 3, where a larger interval is implemented in the latter case as the chosen parameters showed little variability over the shorter time step **(a-c):** $N_{ice}$; **(d-f):** $N_{ice>100\mu m}$; and **(g-i):** $Q_{liq}$. Observed $N_{ice}$ data from noted shattering event (Young et al., 2016a) are excluded in panels **c** and **f**, so that only primary contributions of ice are considered.

### 4.2.3 Case 3: Ocean

Figure 6**(c, f, i)** shows that there is more variation between the parameterisations over the ocean (case 3). C86 causes cloud glaciation at approximately 17 h. Liquid water is simulated at cloud top until this point. D10 produces a mixed-phase cloud layer for the full 24 h duration of the run. Again, more liquid and less ice is modelled using ACC (Table 3). D10×0.1 produces
5  peak $N_{ice}$ values that are almost a factor of 2 lower than observed for case 3 (0.37 L$^{-1}$), allowing the greatest peak $Q_{liq}$ to form





out of the five parameterisations considered ($0.38\,\mathrm{g\,kg^{-1}}$, Fig. 8**i**, Table 3). $Q_{liq}$ is high with comparison to the ACCACIA observations (Table 2); however, $N_{ice}$ is in better agreement than the D10 simulations in this case. In contrast, D10×10 causes rapid glaciating events to occur (Fig. 8**c**). These repeat every ∼3 h of the model simulation. Little liquid water is produced throughout (∼$0.1\,\mathrm{g\,kg^{-1}}$); however, small increases are modelled alongside the glaciating bursts.

Substantial differences can be identified between the three main parameterisations considered. At 7 h, C86 produces the highest $N_{ice}$, with D10 producing the second greatest and ACC producing the least (Fig. 7**c**). D10 and C86 overpredict $N_{ice>100\mu m}$, as with cases 1 and 2. ACC provides the best agreement with the mean $N_{ice}$ observed, simulating approximately $0.4\,\mathrm{L^{-1}}$. ACC also produces a comparable $N_{ice>100\mu m}$ (∼$0.2\,\mathrm{L^{-1}}$, Fig. 7**f**). A more complex picture occurs in Fig. 7**i**: $Q_{liq}$ is at its greatest at the chosen time step (7 h, Fig. S8), therefore the ±4 h variability illustrated is always less than the mean modelled profile shown

using each parameterisation. As with case 1, C86 and D10 underestimate $Q_{liq}$, and ACC performs well, with comparison to the observations.

The steady increase of IWP and LWP seen in cases 1 and 2 is not modelled in case 3: all simulations produce a decreasing LWP with time, whilst the majority also produce a decreasing IWP (Fig. 9**c, f**). A consistent IWP and steadily decreasing LWP are modelled with ACC. The rapid glaciating events modelled with D10×10 (shown in Fig. 8**c**) can again be seen in the IWP,

with a maximum value of nearly $25\,\mathrm{g\,m^{-2}}$ attained at approximately 14 h (Fig. 9**f**). The LWP is zero for the majority of this simulation; however, a small amount of liquid also forms at 14 h. As with case 2, D10 and C86 produce similar IWP and LWPs in case 3 for the majority of the simulations; however, these diverge at approximately 17 h when the C86 case glaciates (Fig. 9**c, f**).

During the D10 simulation, peculiar trends form in both the LWP and IWP traces at approximately 19 h. Peaks and troughs

in the IWP trace correspond with peaks in the LWP at approximately 20 h and 22 h. To investigate these LWP and IWP trends further, Fig. 10 shows X-Y planar views of each simulated parameterisation at 21 h: LWP and IWP are total integrated values over the full height of the domain, and W is chosen at approximately cloud top (1500 m). Little variation can be seen in D10×10 (Fig. 10**a**) and C86 (Fig. 10**b**) at this time as $N_{ice}$ and $Q_{liq}$ have dissipated and not reformed yet. Co-located hot spots of IWP, LWP, and vertical velocity can be seen in the D10 simulation (Fig. 10**c**). Strong updraughts are modelled in close vicinity to

enhanced downdraughts. Regions of high LWP or IWP are not seen in the ACC case (Fig. 10**d**); however, similar activity can be identified in the D10×0.1 (Fig. 10**e**) case. This structure is most visible in the LWP as little ice is simulated.

The parameterisations represented in Fig.10**(c, d, e)** were considered further: the D10 case produces the most ice and least liquid of the three, with D10×0.1 vice versa. Hot-spots of LWP, IWP, and W form with D10, but not with ACC. Defined structure can be seen in the LWP of D10×0.1, and this shape mirrors a region of isolated downdraughts (Fig.10**e**). These fea-

tures may be linked to precipitation from the simulated cloud, and Fig. 11 shows the solid (snow + graupel) and liquid (rain) precipitation modelled in the D10, ACC, and D10×0.1 simulations for case 3. With D10, a greater number concentration of solid precipitation (up to $1\,\mathrm{L^{-1}}$) is modelled than in the ACC ($0.29\,\mathrm{L^{-1}}$) or D10×0.1 ($0.17\,\mathrm{L^{-1}}$) simulations. Similarly, significantly more rain is modelled (up to $27\,\mathrm{L^{-1}}$) in the D10×0.1 simulation in comparison to ACC ($17\,\mathrm{L^{-1}}$) or D10 ($12\,\mathrm{L^{-1}}$). With comparison to D10 and D10×0.1, ACC produces less solid and less liquid precipitation respectively. Precipitation modelled

during cases 1 and 2 are shown in the Supplement (Fig. S9).





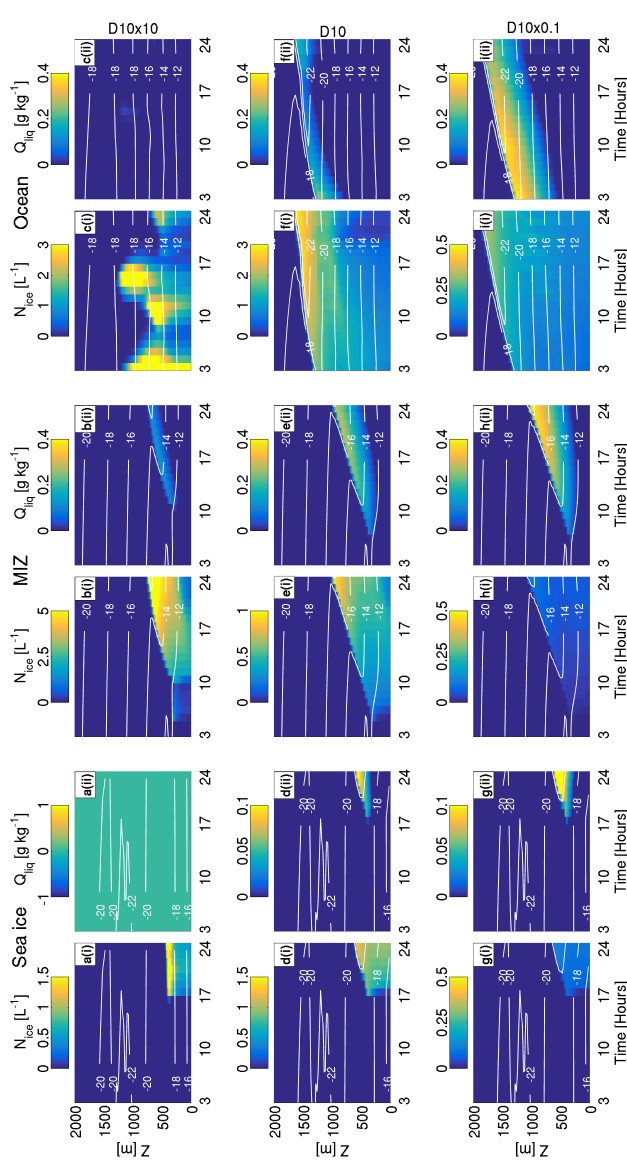

**Figure 8.** Sensitivity of cloud structure to ice crystal number. **(a-c):** D10×10, **(d-f):** D10, **(g-i):** D10×0.1. As previous, $N_{ice}$ and $Q_{liq}$ are shown, and columns indicate sea ice, MIZ, and ocean from left to right. Run length 24 hours. Temperature (°C) contours are overlaid in white. Note changing colour bars for each subfigure.





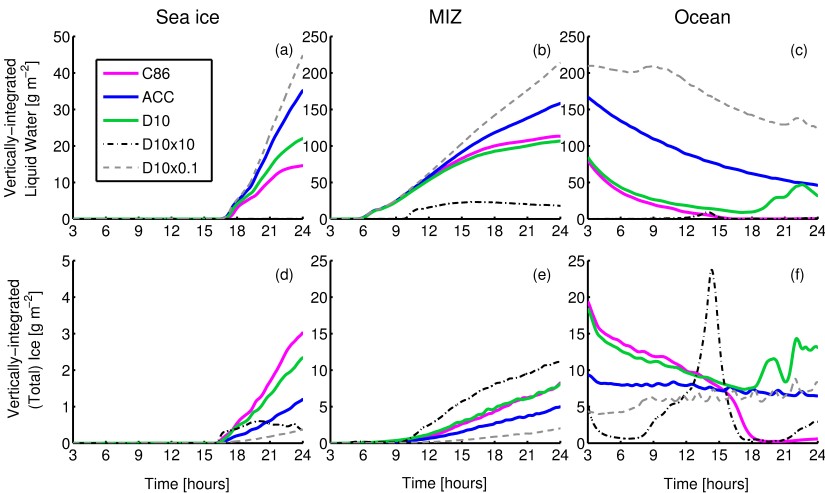

**Figure 9.** Vertically-integrated liquid (**a-c**) and ice water paths (**d-f**) for the sea ice, MIZ, and ocean cases when implementing each of the C86, ACC, D10, D10×10, and D10×0.1 parameterisations under water-saturated conditions.

## 5  Discussion

### 5.1  Ice nucleation at water-saturation: Cooper (1986)

Using in situ observations for reference, we have shown that ice nucleation under water-saturated conditions allows mixed-phase conditions to be modelled over the sea ice, marginal ice zone (MIZ), and ocean. Using C86 deposition-condensation
5  freezing in the Morrison microphysics scheme (Morrison et al., 2005) as a control for each case, the mixed-phase conditions observed over the MIZ (case 2) and the ocean (case 3) are captured by the model; however, no liquid is modelled over the sea ice (case 1). Cases 2 and 3 impose surface fluxes from the simulated ocean surface below; fluxes which induce turbulence in the modelled clouds. The lack of strong surface sensible and latent heat fluxes in case 1 restricts the formation of liquid water in the model as the second imposed criterion of ice supersaturation ($S_i > 1.08$) is attained first. This modelled microphysics
10  is unrepresentative of the observations during case 1. It is unlikely that the nucleation mechanisms involved in these clouds would differ substantially between the sea ice, MIZ, and ocean. Therefore, we suggest that deposition-condensation freezing is ineffective at ubiquitously reproducing Arctic MPS over the range of surfaces possible.





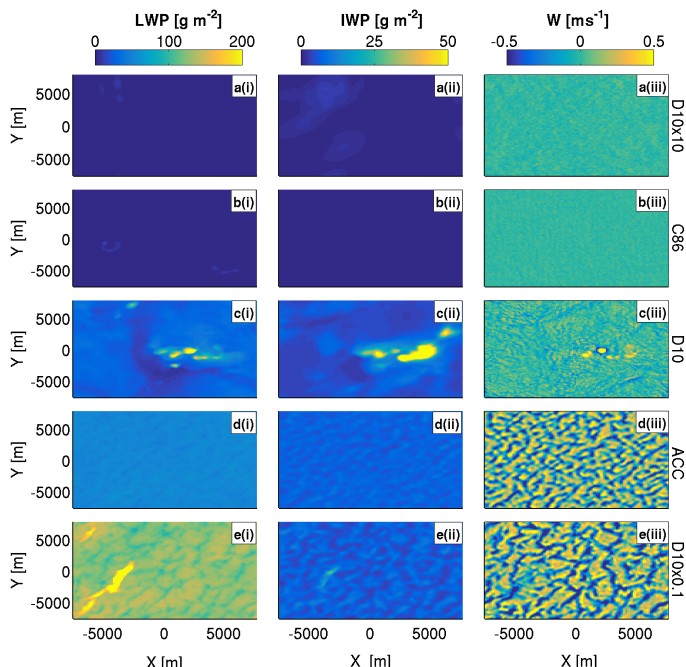

**Figure 10.** Liquid and ice water path (**first** (**i**) and **second** (**ii**) columns) and vertical velocity at approximately 1500 m (**third** (**iii**) column) for each of the five ice nucleation parameterisation scenarios in the ocean case. Planar X-Y slices are shown at 21 h. Runs are arranged such that the simulation which produced the most ice (D10×10, **a**) is on the top row, and that which produced the least ice (D10×0.1, **e**) is on the bottom row. Colour bar at the top of each column corresponds to data in that column only.

## 5.2 Relationship with predicted INPs: DeMott et al. (2010)

Of the two established parameterisations considered (Cooper 1986 and DeMott et al. 2010), D10 produces the best agreement with the observed ice and liquid in all cases. In particular, it reproduces the low ice number concentrations observed during case 2.

5   D10 predicts the number of INPs – not ice crystals – active at a given temperature, $T_K$. Though reasonable agreement is found with observations, D10 still produces too many ice crystals in each case (Fig. 6**d, e, f**). D10 predicts approximately double and quadruple the number of ice crystals observed at the respective CTTs in cases 1 and 3 (Tables 1 and 2). Young et al. (2016b) found a large fraction of super-micron sea salt particles over the sea ice (case 1) and below the MIZ cloud (case 2). No filter data were available for the ocean case (case 3); however, it can be assumed that a similar fraction of these aerosol particles



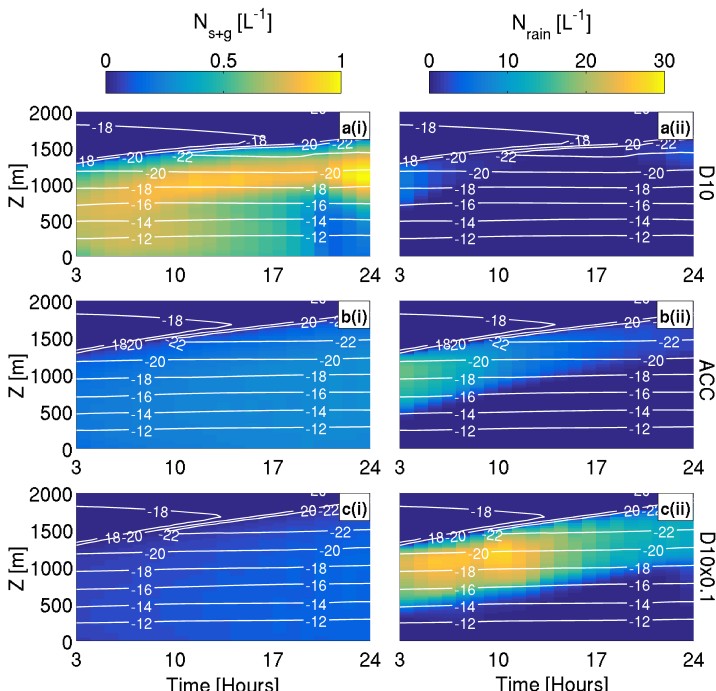

**Figure 11.** Summed snow and graupel number concentrations ($N_{s+g}$, **(i)**) and rain number concentration ($N_{rain}$, **(ii)**) using **(a)** D10 , **(b)** ACC, and **(c)** D10×0.1 during case 3. Run length 24 hours. Temperature (°C) contours are overlaid in white. Colour bar at the top of each column corresponds to data in that column only.

may also be sea salt, given they were found upstream over the sea ice under the same meteorological conditions (Young et al., 2016a). Given these results, it is not surprising that D10 overestimates the quantity of super-micron INPs available to nucleate ice in these conditions, as sea salt is an inefficient INP and constitutes a large fraction of $n_{aer, >0.5\,\mu m}$.

Additionally, an approximation of D10 was applied. The average aerosol number concentration ($0.5 < D_P \leq 1.6\,\mu m$, DeMott et al., 2010) in each case was used to evaluate Eq. 2 to give a temperature-dependent function. This idealised scenario would only be representative of a region where the aerosol particle number concentration was being replenished and INPs were not depleted. Such replenishment is likely unrepresentative of the Arctic, as there are few in situ sources of INP in this region. Additionally, a constant input of aerosol particle number concentration was used in Eq. 2, irrespective of altitude in the model; therefore, spatial variability of INPs in the boundary layer is not represented. Particle number concentrations typically decrease





with altitude away from local surface sources; therefore, this approximation of vertical homogeneity may also be positively influencing the number concentration of ice crystals predicted by D10.

### 5.3 ACCACIA observational fit: ACC

For the three case studies considered, the ACC relationship performs well. Cloud temperatures are colder and cloud top is higher than in the C86, D10, or D10×10 simulations, due to strong radiative cooling from the heightened $Q_{liq}$. The liquid-dominated clouds modelled display enhanced cloud-driven convection across the full domain (Figs. 10, S2-S4). Only the D10×0.1 simulations produce colder temperatures and a higher cloud top. ACC produces the best microphysical agreement with observations for cases 1 and 3 (Fig. 7**g, i**); however, too much $Q_{liq}$ is modelled in case 2 (Fig. 7**h**). Too few ice crystals are modelled to sufficiently deplete the liquid phase via the WBF mechanism. Ice crystal habits are not explicitly resolved in the microphysics scheme, which could influence the modelled $Q_{liq}$. Habits which undergo efficient vapour growth (e.g. stellar dendrites or sector plates, Mason, 1993) would allow increased ice mass to be modelled, with a consistent $N_{ice}$ and a suppressed $Q_{liq}$.

The ACC relationship was derived from 2DS ice number concentration data from five springtime ACCACIA flights. The small sample size restricted the range over which a relationship could be established. Ice observations between 252 K and 265 K were collected; therefore, the dependence of ice number concentration on temperatures outwith this range could not be established. Further observations in this temperature range could allow this relationship to be validated and potentially extended further; however, based on these ACCACIA data, this curve is not applicable beyond $252\,\text{K} < T_k < 265\,\text{K}$. Temperatures colder than this limit are modelled in case 3 due to increasing cloud top height and strong radiative cooling; therefore, these results must be interpreted with caution.

### 5.4 Ice number sensitivity

#### 5.4.1 Cloud microphysics

As shown by previous studies (Harrington and Olsson, 2001; Morrison et al., 2011; Ovchinnikov et al., 2011, amongst others), the microphysical structure of Arctic MPS is highly sensitive to ice crystal number. Greater ice number concentrations enhance the efficiency of the WBF process – leading to the depletion of liquid water within the cloud – whilst lower number concentrations allow liquid droplets to persist under moderate vertical motions. D10 sensitivity tests for cases 1 and 2 behave as would be expected: D10×0.1 produces significantly less ice, allowing liquid to dominate and cloud top height to increase, whilst D10×10 produces high ice number concentrations which glaciate case 1 and strongly suppress the liquid of case 2 (Fig. 8). Mixed-phase conditions are maintained in case 2; however, $N_{ice}$ is much larger than observed using this parameterisation (Fig. 4, Table 2). Additionally, in case 3, D10×10 causes rapid glaciating events to occur.

To compare between our water-saturated simulations, we define two stages of cloud evolution: a formation phase, characterised by an increasing LWP, and a decaying phase, with a decreasing LWP. In cases 1 and 2, each parameterisation causes the simulated clouds to remain in the formation phase by the end of each run (Fig. 9**c, f**). Both the LWP and IWP typically increase



during these simulations. The LWP of case 2 begins to plateau towards 24 h in these cases, indicating the possible start of the decaying phase. Case 3 attained this phase immediately, as shown by the decreasing LWPs modelled.

Modelled LWPs and IWPs are smaller in case 1 than in both cases 2 and 3. Case 1 imposed negligible surface fluxes; therefore, cloud dynamics was driven primarily by longwave radiative cooling (similarly to Ovchinnikov et al., 2011). In the observations (Young et al., 2016a), a lack of strong turbulent motions within this cloud layer caused a suppressed LWMR in the vicinity of moderate ice number concentrations. The LEM reproduces these conditions well in the absence of strong surface fluxes (sensible heat fluxes of $1\,\mathrm{W\,m^{-2}}$ imposed). The ocean-surface cases (2 and 3) implement strong surface fluxes, allowing turbulent motions to sustain a greater $Q_{liq}$ within the mixed-phase cloud layer (Morrison et al., 2008).

Cloud top height clearly increases with model time in cases 1 and 2, and more subtly in case 3. Large-scale subsidence, which would act to suppress cloud top ascent, was not imposed in these simulations. This increasing cloud top was observed by Young et al. (2016a) over the transition from sea ice to ocean; therefore, the modelled cloud structure is in good agreement with observations without large-scale subsidence imposed. However, the temperatures simulated in case 2 (Figs. 6, 8**b, e, h**) are colder than observed (Table 1). Despite this, $N_{ice}$ modelled with the temperature-dependent parameterisations considered is in reasonable agreement with the observations (Fig. 7**b**). Case 2 occurred on a different day to cases 1 and 3; therefore, different synoptic conditions were influencing the sampled cloud systems. Increasing the modelled large-scale subsidence acts to increase the modelled temperatures (not shown, Fig. S10); however, a substantial subsidence would be required to match the observations. Given that imposing large-scale subsidence increases the temperature and suppresses $Q_{liq}$, without greatly affecting $N_{ice}$, we suggest that a greater imposed subsidence may improve the agreement with the observations in case 2.

Cloud top reaches higher altitudes in the ACC and D10×0.1 simulations – across all surfaces – compared to D10, C86, and D10×10, due to a greater liquid water content; as more liquid forms from the vapour field, more heat is released, pushing the cloud top higher. These liquid-dominated cases are also shown to experience enhanced convection across the full domain in case 3 (Fig. 10). With increased cloud top height, enhanced radiative and evaporative cooling enforce downdraughts whilst increased latent heat release from droplet formation and growth strengthens updraughts. In the C86, D10, and D10×10 simulations, a greater $N_{ice}$ suppresses efficient droplet growth, latent heat release, cloud top ascent, and strong radiative cooling through the WBF mechanism. This finding is in agreement with Harrington and Olsson (2001), who showed that high $N_{ice}$ produced weaker BL convection and a shallower BL, whilst liquid-dominated mixed-phase clouds promote a higher cloud top and deeper BL.

### 5.4.2 Cloud glaciation or break up

Over the ocean (case 3), C86 leads to cloud glaciation when freezing is implemented under both deposition-condensation (Fig. 5**c**) and water-saturated (Fig. 6**c**) conditions. This cloud glaciation is tied to the number of ice crystals produced: over the temperature range shown in Fig. 1, D10×10 and C86 typically produce the most ice, and so rapid ice formation is simulated once the onset thresholds are reached. This suppresses the liquid phase within the cloud layers, either immediately (D10×10) or after an accumulation period (C86). However, D10 produces a similar $N_{ice}$ ($2.07\,\mathrm{L^{-1}}$) to C86 ($2.42\,\mathrm{L^{-1}}$, Table 1) at the CTTs considered. This subtle difference in predicted ice number allows the D10 cloud to persist, whilst the C86 cloud glaciates.



While D10 produces a persistent mixed-phase cloud for the full duration, peculiar trends appear at times >20 h. Figure 9 shows the development of peaks and troughs in the IWP, with corresponding peaks in the LWP, after this time. From Fig. 10**c**, localised hot spots of LWP, IWP, and vertical velocity can be seen. These localised regions of increased ice and/or liquid result from isolated convective cells within the cloud. The formation of these cells forces the cloud top higher (Fig. 6**f**), with renewed liquid and ice formation. Similar structures can be seen in the D10×0.1 simulations (Fig. 10**e**); however, these appear mostly in the LWP field and have an elongated, banded shape in comparison to the compact, almost circular, structures which evolve in D10. These localised regions of enhanced convection can be linked to increased precipitation (Fig. 11). Specifically, increased solid (snow + graupel) precipitation is modelled using D10, whilst increased liquid (rain) precipitation is modelled in D10×0.1.

The formation of convective cells in the ocean case mirrors cold air outbreak observations: as cold air moves from over the sea ice to the ocean, the boundary layer becomes thermodynamically unstable, allowing temperature perturbations to cause strong positive feedbacks on the cloud structure. Mixed-phase clouds are sustained by moderate vertical motions (e.g. Shupe et al., 2008a, b), driven by latent heating from hydrometeor growth within the cloud and radiative cooling at cloud top (Pinto, 1998; Harrington and Olsson, 2001). At the cold temperatures considered (approximately -20 °C), ice grows favourably by vapour growth in the vicinity of liquid droplets and, given a high enough $N_{ice}$, updraughts are enhanced through latent heat release. With enforced updraughts, water supersaturations are sustained, more cloud droplets form, and cloud top is forced to higher altitudes. With more liquid and a higher cloud top, enhanced radiative cooling strengthens downdraughts adjacent to the updraught columns. With a deeper cloud layer, precipitation can form by an increased likelihood of collision-coalescence of droplets, or ice crystal growth and aggregation, within downdraughts. The formation of precipitation warms and dries the cloud, reinforcing the updraughts and recycling the process. In the D10 ocean case – with high ice number concentrations, but not high enough for glaciation – the accumulation of $N_{ice}$ promotes this pathway, with the development of precipitation being the key factor in the localised, runaway convection that occurs.

With precipitation as snow or rain, the $Q_{liq}$ is depleted from the cloud layer. The D10 case produces high number concentrations of snow, which depletes $Q_{liq}$ efficiently. Once the convective activity starts in this case, the cloud liquid is depleted; however, it is also partially restored through sustained supersaturations in the strong updraughts. In the D10×0.1 case, the $Q_{liq}$ depletion is slower as rain is less efficient at removing droplets than snow. Both of these precipitation pathways would therefore likely lead to cloud break up if the simulation time was extended further.

Given the two pathways of precipitation identified by Fig. 11, a question arose: do these structures form as a result of the functional form of D10, or are they related simply to ice number? ACC produced an $N_{ice}$ between D10 and D10×0.1, and no heterogeneous structures were observed. Therefore, to address this question, D10×0.5 was imposed in the LEM (see Fig. 1). For comparison with Table 1, D10×0.5 predicts 1.04 L$^{-1}$ at the CTT. Figure 12 illustrates modelled $N_{ice}$ and $Q_{liq}$ for the D10, D10×0.5, and D10×0.1 simulations over the ocean. LWP and IWP modelled at 21 h are also shown. D10×0.5 produces less ice than D10 and less liquid than D10×0.1: this simulation behaves as expected to also give the microphysical mid-point between the D10 and D10×0.1 scenarios. Therefore, the modelled cloud persistence and stability is not just a feature of ACC. A homogeneous cloud structure is modelled with D10×0.5 and the localised hot-spots of the D10 and D10×0.1 cases is not seen. Such hot-spots never form in the D10×0.5 simulation. Modelled precipitation (Fig. S11 in the Supplement) using this



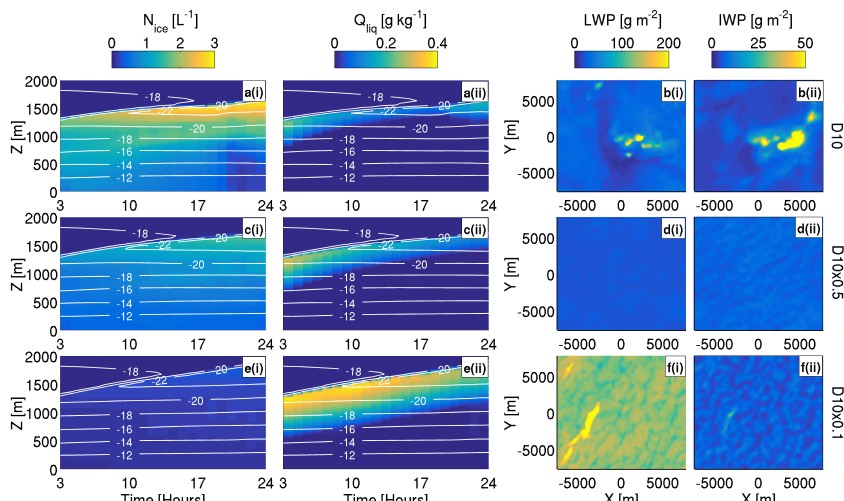

**Figure 12.** $N_{ice}$, $Q_{liq}$, LWP, and IWP modelled in the D10 (**a,b**), D10×0.5 (**c-d**), and D10×0.1 (**e-f**) simulations over the ocean (case 3). $N_{ice}$ and $Q_{liq}$ are shown in the **first** and **second** columns, plotted against time and altitude as previously. LWP and IWP (**third** and **fourth** columns) are X-Y planar views at 21 h, as also shown in Fig. 10. (**a, c, e**): Temperature (°C) contours overlaid in white. Colour bar at the top of each column corresponds to data in that column only.

parameterisation is significantly less than D10 (snow + graupel) and D10×0.1 (rain), and the simulated cloud persists for the full 24 h duration with no break up.

Additionally, a larger domain size was imposed to test if these convective cells were related to the imposed cyclical boundary conditions: both similar structures and LWP / IWP trends formed (not shown, Figs. S12, S13), suggesting these convective cells

5   are not simply a result of the domain configuration. Within the time scales imposed in this study (24 h), these cells are only observed over the ocean (case 3). Given more time (33 h), case 2 also develops convective cells and precipitation when D10 and D10×0.1 are imposed (not shown, Figs. S14– S17). Therefore, we conclude that – in two of the ACCACIA cases considered, which occurred on different days, under different synoptic conditions, with different air mass histories (Young et al., 2016b) – model simulations using the D10 ice nucleation parameterisation can produce localised cellular structure within the mixed-

10  phase cloud layer, given enough time to do so.

Here, the development of appreciable precipitation is particularly sensitive to ice number. ACC and D10×0.5 maintain mixed-phase conditions for 24 h over the ocean, with no cell development (Fig. 12), suggesting there is a "sweet spot" for $N_{ice}$ in this case. Glaciation occurs with C86, persistence is achieved with D10×0.5 and ACC, and convective cells form in D10 and D10×0.1: it is unclear which representation is correct in this environment, as observations do show the development of

15  roll convection in cold air outbreak scenarios as the cold air masses move over the warm ocean (e.g. Hartmann et al., 1997).





Additionally, snow precipitation was observed by Young et al. (2016a) in this case. It cannot be stated whether the time scales of convection development modelled here are good representations of this phenomenon.

### 5.4.3 Cloud persistence

Mixed-phase conditions are sustained for at least 8 h in all three cases when imposing the three main parameterisations; ACC,
D10, and C86. By additionally considering the sensitivity tests (C86, D10, ACC, D10×10, and D10×0.1), we can suggest limitations of $N_{ice}$ which maximise cloud persistence in each case, based on the predictions at the cloud top temperature (Table 1). Over the sea ice (case 1), ACC compares particularly well (producing $0.51 \, L^{-1}$ at the CTT) and produces a comparable $Q_{liq}$ (Fig. 7**a, g**). D10 only marginally overpredicts $N_{ice}$ ($1.31 \, L^{-1}$, Table 1). D10×0.1 produces too few ice crystals and simulated $Q_{liq}$ is too high with comparison to the observations (Table 3). C86 predicts too much ice ($2.03 \, L^{-1}$, Table 1), peaking at $2.32 \, L^{-1}$ when implemented in the model (Table 3). Given these results, optimal mixed-phase cloud persistence, and comparable microphysics, are simulated with $0.51 \, L^{-1} < N_{ice(CTT)} < 1.31 \, L^{-1}$ over the sea ice, where the upper limit is more than twice the mean observed throughout the cloud layer ($0.47 \pm 0.86 \, L^{-1}$, Table 2), but is still within $1\sigma$.

Over the MIZ (case 2), all parameterisations produce a mixed-phase, sustained cloud layer. Without observations to compare to, it may not have been possible to identify which has a more realistic grounding. However, we have shown that C86 and D10 perform similarly and produce marginally higher ice number concentrations than are observed (Fig. 7**b**). These relationships predict ice/INP number concentrations of $0.23 \, L^{-1}$ / $0.34 \, L^{-1}$ respectively at the CTT of case 2. $Q_{liq}$ also agrees well with observations when implementing these parameterisations. However, both C86 and D10 overpredict $N_{ice>100\mu m}$, suggesting the majority of modelled ice is growing too efficiently. This is likely representative of this case, as the warm, sub-zero cloud temperatures (-13 °C) would not promote efficient ice crystal growth. ACC also produces a sustained, mixed-phase cloud layer in case 2; however, a significantly greater $Q_{liq}$ is modelled than is observed ($0.22 \, g \, kg^{-1}$, compared with $0.07 \, g \, kg^{-1}$, at 700 m in Fig. 7**h**). This suggests that the simulated ice number concentration is not sufficient enough to suppress the formation of liquid with this relationship. Optimal mixed-phase cloud persistence and comparable microphysical structure is modelled when $0.23 \, L^{-1} < N_{ice(CTT)} < 0.34 \, L^{-1}$ over the MIZ, where the upper limit is in good agreement with the mean observed and the lower limit is within one standard deviation ($0.35 \pm 0.20$, Table 2).

Over the ocean (case 3), strong sensitivities to $N_{ice}$ emerge. D10×10 simulates a high $N_{ice}$; therefore rapid, repeating glaciating events occur. This is not representative of the persistent, mixed-phase MPS of interest. C86 allows a mixed-phase cloud layer to form for some time, approximately 17 h, after which it glaciates due to accumulated ice concentrations. This glaciating event does not occur with D10, even though only $\sim 0.4 \, L^{-1}$ less ice is predicted at the CTT. Both C86 and D10 do not reproduce the observed cloud liquid well (Fig. 7**i**). D10×0.1 produces reasonable agreement with the $N_{ice}$ and $Q_{liq}$ observations at 7 h (Fig. S5); however, the rapidly increasing cloud top height and $Q_{liq}$ with time are not also representative of the observations. As discussed in Sect. 5.4.1, large-scale subsidence may help to constrain these properties. ACC provides good agreement with both $N_{ice}$ and $Q_{liq}$ – when not considering the shattering event at cloud base (Fig. 4**c**) – where $0.54 \, L^{-1}$ is predicted at the case 3 CTT. This is in very good agreement with the ice number concentration observed ($0.55 \pm 0.95 \, L^{-1}$, Table 2). The $N_{ice}$ values predicted in D10 and D10×0.1 produced a microphysical structure with enhanced precipitation,



which may lead to cloud break up after time. Steady mixed-phase conditions were only simulated when implementing ACC and D10×0.5. Therefore, to simulate a consistent cloud layer over the ocean in case 3, $0.54^{-1} < N_{ice(CTT)} < 1.04\,L^{-1}$ is required.

From these three cases, it is clear that small differences in the predicted $N_{ice}$ can produce significant microphysical impacts on the modelled clouds. The best prediction of $N_{ice}$ for each case is different; however, they are of a similar order of magnitude and vary only a little between each case. Case 2 requires the least $N_{ice}$ due to the comparatively warmer CTT (-12.7 °C), whereas cases 1 and 3 – with similar CTTs (approximately -20 °C) – require $N_{ice}$ over a similar range (approximately $0.5\,L^{-1}$ to $1.3\,L^{-1}$) to produce a sustained, mixed-phase cloud layer with $N_{ice}$ and $Q_{liq}$ in approximate agreement with in situ observations. These limitations are based upon the parameterisations chosen in this study (C86, D10, ACC, D10×0.1, and D10×10); therefore, further work should be conducted to test other relationships and constrain the identified limitations in each case. These results are in accordance with Ovchinnikov et al. (2011), whose modelled springtime Arctic MPS glaciated when an ice number concentration of $2\,L^{-1}$ was imposed, whilst $0.5\,L^{-1}$ produced mixed-phase conditions with both consistent LWP and IWPs attained after ∼3.5 h. Given these are idealised simulations (with constant SW radiation and no INP depletion), the ability of the model to simulate realistic conditions should be inferred with caution: results from this study can simply conclude that small increases in the modelled ice crystal number concentration can cause persistent mixed-phase clouds to glaciate.

## 6 Conclusions

In this study, we have used large eddy simulations to investigate the microphysical sensitivity of Arctic mixed-phase clouds to primary ice number concentrations and surface conditions. The Large Eddy Model (LEM, UK Met Office, Gray et al., 2001) was used to simulate cloud structure and evolution over the sea ice, marginal ice zone (MIZ), and ocean. Aircraft observations of cloud microphysics from the Aerosol-Cloud Coupling and Climate Interactions in the Arctic (ACCACIA) campaign were used as a guide to indicate which simulations gave the most realistic microphysical representation. We used two primary ice nucleation parameterisations (Cooper, 1986; DeMott et al., 2010, abbreviated to C86 and D10 respectively), one derived from ACCACIA observations (ACC, Eq. 3), and an upper and lower sensitivity test (D10×10 and D10×0.1) to produce ice crystal number concentrations within the modelled clouds.

Three main sensitivities arise from the three considered cases.

- C86 cannot reproduce the sea ice cloud (case 1) under the conditions commonly used in the Weather Research and Forecasting (WRF) model with the Morrison et al. (2005) microphysics scheme (Fig. 5). However, these criteria do allow for a mixed-phase layer to form in cases 2 and 3, when the ocean provides strong sensible heat fluxes to the BL. This result demonstrates that deposition ice nucleation is not wholly representative of ice nucleation in the Arctic springtime clouds observed during the ACCACIA campaign. Ice nucleation in water-saturated conditions must be implemented to create a mixed-phase cloud layer in our three considered cases (Fig. 6).





– Warm supercooled mixed-phase clouds over the MIZ can be modelled to reasonable accuracy by using the C86 and D10 parameterisations (Figs. 6, 7). At the cloud top temperature attained by case 2 (-12.7 °C), the difference between the C86 and D10 parameterisations is small (Fig. 1, Table 1). These parameterisations overpredict the ice number concentrations at the colder temperatures modelled in cases 1 and 3 (approximately -20 °C). ACC is modulated to have a weakened temperature-dependence; therefore, persistent, mixed-phase cloud layers are modelled in all three cases using this relationship.

– Results shown here illustrate that microphysical structure is particularly sensitive to the modelled ice crystal number concentration when simulating clouds over an ocean surface. With marginally too much ice (e.g. 2.43 L$^{-1}$, C86, Table 1), cloud glaciation occurs. Slightly less ice (2.07 L$^{-1}$, D10, Table 1) allows for persistent mixed-phase conditions for some time (approximately 24 h); however, convective cells form with heightened snow precipitation, which may promote cloud break up. Conversely, too much liquid and very few ice crystals (0.21 L$^{-1}$, D10×0.1, Table 1) may also promote cloud break up via precipitation as rain. Case 3 simulations show that there is a "sweet spot" for simulating ice in ocean-based single-layer Arctic MPS (attained by ACC and D10×0.5), where the number concentration of ice is low enough to sustain a reasonable $Q_{liq}$ through vertical motions and high enough to suppress the liquid phase and restrict efficient collision-coalescence and rain formation. In this narrow limit, the influence of the WBF mechanism is depleted. The fact that this "sweet spot" can be attained by halving the D10 prediction of INP number concentration – yet it is overshot with D10×0.1 – illustrates just how sensitive the cloud structure is to the ice phase. Therefore, we suggest that the method of parameterising the ice number concentration in bulk microphysical models is very important, as small differences in the predicted ice concentration can have substantial effects on the microphysical structure and lifetime of Arctic MPS.

These idealised simulations assume an infinite source of INPs to the modelled clouds; here, INP are not depleted by activation or precipitation. An infinite source of INPs is likely unrepresentative of the Arctic environment (Pinto, 1998), as there are few in situ sources of INPs (e.g. mineral dusts, Murray et al., 2012). Although Young et al. (2016b) identified mineral dusts during all flights of the ACCACIA campaign, further work should include prognosing INPs in such simulations to investigate how their depletion could affect the microphysical structure of these clouds. Several studies have previously identified INP depletion as an important process to represent in modelling Arctic MPS (Harrington et al., 1999; Harrington and Olsson, 2001, amongst others). Additionally, the Morrison et al. (2005) microphysics scheme has been used for its detailed representation of microphysical interactions, such as ice aggregation and growth, but it can be utilised further to represent aerosol particle properties. Size distributions can be prescribed; therefore, the D10 parameterisation could be developed to give a spatially-dependent INP prediction based on aerosol particle observations, likely leading to a more comprehensive treatment of INP variability throughout the domain.



*Acknowledgements.* This work was funded as part of the ACCACIA campaign (grant NE/I028696/1) by the National Environment Research
Council (NERC). G. Young was supported by a NERC PhD studentship. We would like to thank A. Hill and B. Shipway for advising on the
microphysics scheme. Airborne data were obtained using the BAe-146-301 Atmospheric Research Aircraft [ARA] flown by Directflight Ltd
and managed by the Facility for Airborne Atmospheric Measurements [FAAM], which is a joint entity of the Natural Environment Research
5   Council [NERC] and the Met Office. Sea ice data were obtained from the National Snow and Ice Data Centre (NSIDC).





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
