# Peer review of "Microphysical sensitivity of coupled springtime Arctic stratocumulus to modelled primary ice over the ice pack, marginal ice, and ocean"

_Atmospheric Chemistry and Physics, 2016_

## Referee Comment (RC1) · Anonymous Referee #1 · 15 Nov 2016

Review of "Microphysical sensitivity of coupled springtime Arctic stratocumulus to modelled primary ice over the ice pack, marginal ice and ocean" by Young et al.

This paper uses LES simulations of mixed phase Arctic clouds in order to determine the sensitivity of the microphysical structure and lifetime of single-layer mixed phase stratocumulus clouds in the Arctic to three different ice nucleation parameterizations. Overall, I think this is a excellent paper that would be a great fit for ACP, and honestly all I think the authors need to do is to improve the presentation in some of the figures. The uncertainties in the model simulations and data are very thoroughly discussed, the procedures they used are well documented, and I think that it is interesting to find that small uncertainties in Nice can lead to drastically different simulations, which further

adds to our need to better characterize the microphysical observations of mixed phase stratocumulus.

I have some minor comments that would improve the paper. In general, I do think Figures 5,6, and 8 do need to be more readable, as many of the sub panels are quite small and can be difficult to read at points.

I also think that Figures 7a,b,c are not needed as the conclusions reached by those figures could be misleading since you are comparing modeled Nice to observed Nice>100, which are not the same quantity since they cover different size ranges of particles. Particles with sizes less than 100 microns can vastly outnumber those at larger sizes, so it is crucial to make sure that the number concentrations being compared cover the same size range.

Secondary ice production processes also occur in Arctic mixed phase stratocumulus, such as in Rangno and Hobbs 2001. I think it would be a benefit to mention them in the introduction when you introduce primary nucleation. You can then mention that the focus of your study is on primary nucleation rather than secondary production.

---

## Referee Comment (RC2) · Anonymous Referee #2 · 22 Nov 2016

The authors simulate three cases of mixed-phase cloud near Svalbard, based on observations conducted during March 23 and 29 of 2013 as part of the ACCACIA campaign. The focus of this work is on how ice nucleation parameterizations influence results from the UKMO LEM model in comparison with the observations. The authors use three basic parameterizations: Copper 1986; DeMott 2010; another empirical parameterization (ACC) based on the observations they evaluate against. They also evaluate some extremes of the D10 parameterization and the D10*0.1 is one of the better performers for the ocean case. Overall, D10 and C86 produce more ice, which leaves less liquid water, and ACC produces less ice and more liquid water. In general, ACC compares better with the number concentrations of measured ice particles larger

than 100 um, where the latter is used to put the observations and simulations on the same footing. It is perhaps expected and unfair to say ACC performs better, since it is based on the observations being compared against, but it provides an important perspective on the number concentrations of ice particles. The authors identify a "sweet spot" in the number concentration of ice particles, represented by either the ACC or D10*0.5 parameterizations, at which the balance of liquid and ice in these cold clouds over ocean is able to maintain a persistent mixed phase without glaciation or generating too much convection. In general, the study points to the strong sensitivity of Arctic mixed phase clouds to ice crystal number concentrations. Also, for the three cases studied, the authors show that ice nucleation under water saturated conditions must be implemented.

The paper is long and a little difficult because of the many tests and the bouncing back and forth among figures 6 through 9, but it is otherwise well organized, the discussions are good and the results are interesting and important. The paper is appropriate for ACP.

Specific comments:

1) Section 2.2 - Some clarification of the context of the model and observations is needed. The dropsonde data are used for model initialization. The model then simulates cloud for 24 hours, with the first 3 hours considered as "spin-up". During the 21 hours of simulation, is the model is maintaining the same underlying surface: i.e. ice for case 1, mixed ice and water for case 2 and water for case 3? It would seem that in reality the MIZ cloud may have moved from over the MIZ to open water during the time equivalent to the simulation period. Were the in-situ microphysical observations conducted near the locations of the dropsondes or farther downwind? There appear to be differences in the heights of the clouds between the microphysical measurements and the dropsondes with the microphysics suggesting deeper cloud.

2) Presumably, changes in the numbers and sizes of the cloud droplets will also affect

the WBF process. Why did you use a cloud droplet number concentration of 100/cc, when you have the measurements from the CDP that you could have used?

3) For case 1, how are you sure the liquid phase existed? The CDP is a one dimensional probe. What were the droplet number concentrations?

4) Section 4.1, line 12 – at or below 500 m?

5) Page 12, line 1 – provide a reference for the 2DS statement.

6) Page 14, line 1 – among rather than between.

7) Page 18, lines 1-3 - In figure 7, D10 produces the highest ice numbers for >100 um. Even using the total measured ice numbers, D10 is still high. D10 appears to do relatively well in case 1, and it is closer to the observations than C86 in case 3. Is your reference to case 2 a mistake? If not, please explain how you arrive at this statement. The statement on lines 2-3 of page 19 appears to contradict.

8) Page 19, line 7 – There may be fewer sources, but so little is known about INP in the Arctic that I think this sentence would be better removed.

9) Possibly of interest to the authors, INPs measured at Alert, Nunavut (Mason et al., Atmos. Chem. Phys., 16, 1637–1651, 2016) during spring to early summer vary from 0.2 per litre to 1 per litre for temperatures from 20oC to 25oC, which covers the range of average ice particle number concentrations you report for the three cases.
* * *

---

## Author Comment (AC1) · 14 Feb 2017

* * *
**We would like to thank the reviewers for their useful comments and suggestions which have helped us to improve the manuscript.**

**Reviewers' comments**

Reviewer 1, Reviewer 2, Reviewer 3

**Authors' response is shown in black and bulleted.**
**Quotes from the manuscript are in italics.**

Please note:

1. Some figure numbers have been changed in the revised version of the manuscript. New figure numbers are referenced in any related comments.

2. There has been some substantial restructuring of the article following the comments of the reviewers. This has made the track-changed document very difficult to read. Therefore, we strongly suggest reading the revised manuscript and only consulting the track-changed version for specific alterations. **Any line numbers referenced in the authors' response refer to this revised version of the manuscript.**

**It has been brought to our attention that the dropsondes used to initialise the model were affected by a dry bias, leading to an underestimation of the humidity in the model. Given that we initialise the cloud fields in each simulation from an assumed adiabatic profile, this has had little impact on our conclusions. Nevertheless, we have acknowledged this dry bias in the manuscript, and have provided profiles of the revised data, and an example simulation for justification, in the Supplement. The reasons behind this bias are detailed at the following link:**

https://www.eol.ucar.edu/system/files/software/Aspen/Windows/W7/documents/Tech%20Note%20Dropsonde_Dry_Bias_20160527_v1.3.pdf
* * ** * *
**Reviewer 1:**

Review of "Microphysical sensitivity of coupled springtime Arctic stratocumulus to modelled primary ice over the ice pack, marginal ice and ocean" by Young et al. This paper uses LES simulations of mixed phase Arctic clouds in order to determine the sensitivity of the microphysical structure and lifetime of single-layer mixed phase stratocumulus clouds in the Arctic to three different ice nucleation parameterizations. Overall, I think this is an excellent paper that would be a great fit for ACP, and honestly all I think the authors need to do is to improve the presentation in some of the figures. The uncertainties in the model simulations and data are very thoroughly discussed, the procedures they used are well documented, and I think that it is interesting to find that small uncertainties in Nice can lead to drastically different simulations, which further adds to our need to better characterize the microphysical observations of mixed phase stratocumulus.

I have some minor comments that would improve the paper. In general, I do think Figures 5,6, and 8 do need to be more readable, as many of the sub panels are quite small and can be difficult to read at points.

- We have made Fig. 5 clearer, and split Figs. 6 and 8 into 3 separate figures for each case study. These new figures show each of the 5 implemented parameterisations split over sea ice (Fig. 6), MIZ (Fig. 9), and ocean (Fig. 10).

I also think that Figures 7a,b,c are not needed as the conclusions reached by those figures could be misleading since you are comparing modelled Nice to observed Nice>100, which are not the same quantity since they cover different size ranges of particles. Particles with sizes less than 100 microns can vastly outnumber those at larger sizes, so it is crucial to make sure that the number concentrations being compared cover the same size range.

- Figure 7a-c has been removed following Reviewer 1's concerns.

Secondary ice production processes also occur in Arctic mixed phase stratocumulus, such as in Rangno and Hobbs 2001. I think it would be a benefit to mention them in the introduction when you introduce primary nucleation. You can then mention that the focus of your study is on primary nucleation rather than secondary production.

- The beginning of this paragraph (page 2, lines 20-22) has been updated to reflect Reviewer 1's comment:

  *"Ice crystals may form through primary or secondary processes in Arctic MPS (Rangno and Hobbs, 2001). Here, we focus on primary ice formation as secondary ice production has been shown to be less influential in the springtime MPS we shall consider (e.g. Jackson et al., 2012; Young et al., 2016a)."*
* * ** * *
**Reviewer 2:**

The authors simulate three cases of mixed-phase cloud near Svalbard, based on observations conducted during March 23 and 29 of 2013 as part of the ACCACIA campaign. The focus of this work is on how ice nucleation parameterizations influence results from the UKMO LEM model in comparison with the observations. The authors use three basic parameterizations: Copper 1986; DeMott 2010; another empirical parameterization (ACC) based on the observations they evaluate against. They also evaluate some extremes of the D10 parameterization and the D10*0.1 is one of the better performers for the ocean case. Overall, D10 and C86 produce more ice, which leaves less liquid water, and ACC produces less ice and more liquid water. In general, ACC compares better with the number concentrations of measured ice particles larger than 100 um, where the latter is used to put the observations and simulations on the same footing. It is perhaps expected and unfair to say ACC performs better, since it is based on the observations being compared against, but it provides an important perspective on the number concentrations of ice particles.

- We agree that the "performance" of ACC is poor wording as this would likely agree best with the observations due to its derivation from these. We have updated the language throughout the manuscript to use ACC as an "empirically-derived" comparison to the 2 established parameterisations. However, we do feel it is important that the ACC relationship is able to produce both the persistent mixed-phase conditions in each case and a comparable ice phase to the observations as well.

The authors identify a "sweet spot" in the number concentration of ice particles, represented by either the ACC or D10*0.5 parameterizations, at which the balance of liquid and ice in these cold clouds over ocean is able to maintain a persistent mixed phase without glaciation or generating too much convection. In general, the study points to the strong sensitivity of Arctic mixed phase clouds to ice crystal number concentrations. Also, for the three cases studied, the authors show that ice nucleation under water saturated conditions must be implemented.

The paper is long and a little difficult because of the many tests and the bouncing back and forth among figures 6 through 9, but it is otherwise well organized, the discussions are good and the results are interesting and important. The paper is appropriate for ACP.

- We agree that the paper includes a lot of comparisons and can be complicated to follow. We have made efforts throughout the manuscript to make it easier for the reader to follow our arguments.

**Specific comments:**

1) Section 2.2 - Some clarification of the context of the model and observations is needed. The dropsonde data are used for model initialization. The model then simulates cloud for 24 hours, with the first 3 hours considered as "spin-up". During the 21 hours of simulation, is the model is maintaining the same underlying surface: i.e. ice for case 1, mixed ice and water for case 2 and water for case 3? It would seem that in reality the MIZ cloud may have moved from over the MIZ to open water during the time equivalent to the simulation period.

- For simplicity, the same surface conditions are maintained throughout the full duration of each model run. We acknowledge that this approximation is idealistic and, instead, it is likely that surface temperatures would increase. We have made this clearer in the manuscript (page 4, lines 24-25).

    "*These surface conditions were kept constant throughout each simulation.*"

Were the in-situ microphysical observations conducted near the locations of the dropsondes or farther downwind? There appear to be differences in the heights of the clouds between the microphysical measurements and the dropsondes with the microphysics suggesting deeper cloud.
* * *
- The in-situ observations were conducted geographically close to the dropsonde measurements; however, they were carried out at a later time (case 1 ~ 3 h, case 2 ~ 5 h, case 3 ~ 5 h). We are aware that cloud structure will have changed between these measurement times. As mentioned, this is particularly clear in case 2 where the cloud has deepened between the times of the sonde and in situ measurements. We were limited to the data we had from the campaign, so unfortunately measurements closer in time were not made. We have made it clearer in the manuscript (page 8, lines 6-12) that these measurements are co-located spatially, but not strictly temporally. Additionally, as summarised at the top of this document, a potential dry bias in the dropsonde measurements was brought to our attention after our study was completed. We have addressed this bias in the manuscript, and included details – and additional simulations – in the supplementary material.

  "*These aircraft observations sampled the same geographical location approximately 3-5 hours after the dropsonde measurements; therefore, some evolution in cloud properties between the two data is expected. These dropsonde data were affected by a potential dry bias, as discussed by Young et al. (2016a): corrections were applied after this study was completed, and the revised profiles are shown in Figs. S1 and S2. Whilst the general properties of the modelled clouds are mostly unchanged with these corrections imposed, the development of precipitation is affected (examples shown in Figs. S4, S5). Our conclusions are unaffected by this bias; however, these revised profiles highlight an additional sensitivity to humidity in the three cases considered here (see the Supplementary Material for further details).*"

2) Presumably, changes in the numbers and sizes of the cloud droplets will also affect the WBF process. Why did you use a cloud droplet number concentration of 100/cc, when you have the measurements from the CDP that you could have used?

- We used a consistent cloud droplet number concentration across all simulations to ensure the differences between each case were due to the changes in primary ice number alone. Measured $N_{drop}$ for cases 1, 2, and 3 were $110 \pm 36$ cm$^{-3}$, $141 \pm 66$ cm$^{-3}$, $63 \pm 30$ cm$^{-3}$. Cloud droplet number concentrations for cases 1 and 3 have been previously reported in Young et al., 2016a. We have now reported these values in the manuscript (page 5, lines 6-8).

  "*A prescribed droplet number of 100 cm$^{-3}$, approximated from the measured values of 110 $\pm$ 36 cm$^{-3}$, 141 $\pm$ 66 cm$^{-3}$, and 63 $\pm$ 30 cm$^{-3}$ (Young et al., 2016a) for cases 1, 2, and 3 respectively, is applied in all simulations.*"

3) For case 1, how are you sure the liquid phase existed? The CDP is a one dimensional probe. What were the droplet number concentrations?

- $N_{drop}$ for case 1 was $110 \pm 36$ cm$^{-3}$. These observations have been detailed previously in Young et al., 2016a; therefore, we did not wish to go into detail in order not to encroach on that study. The observations suggest that there was indeed liquid in the case 1 cloud, but the cloud droplets were very small in size (~5µm effective radius). We have now quoted the number concentrations in the manuscript for completeness (see above).

4) Section 4.1, line 12 – at or below 500 m?

- This should read "below 500 m", thank you for highlighting this mistake. We have updated the manuscript accordingly (page 10, line 24).

5) Page 12, line 1 – provide a reference for the 2DS statement.

- References have been included as requested (page 12, line 4 – page 13, line 2):

*"2DS data has poor resolution at small sizes (<80 µm), preventing the particle shape factor from being accurately determined at these sizes (Crosier et al., 2011; Taylor et al., 2016; Young et al., 2016a); therefore, the number concentration of small ice crystals is not a reliable measure with this instrument."*

6) Page 14, line 1 – among rather than between.

- We feel that this change would have changed the meaning of the sentence, and would be incorrect. We acknowledge that this sentence was not clear; therefore, it has been changed to the following (page 12, line 4):

  *"The poor resolution of 2DS data at small sizes (<80um) prevents the particle shape factor from being accurately determined"*

7) Page 18, lines 1-3 - In figure 7, D10 produces the highest ice numbers for >100um. Even using the total measured ice numbers, D10 is still high. D10 appears to do relatively well in case 1, and it is closer to the observations than C86 in case 3. Is your reference to case 2 a mistake? If not, please explain how you arrive at this statement. The statement on lines 2-3 of page 19 appears to contradict.

- The reference to case 2 was not a mistake. When considering the values quoted in Table 3, D10 produces lower N_ice than C86 in every case, and therefore provides better agreement with the low N_ice observed. We realise that this statement is confusing when, in Fig. 7, D10 was shown to give the highest N_ice at the chosen time step.

  Following Reviewer 1's comment, Fig. 7(a-c) has been removed to avoid misleading conclusions. The comparison of large ice (new version of Fig. 7, panel b) also shows that D10 gives the highest, and thus compares the worst, with observations. This suggests that a large fraction of the modelled N_ice by D10 is growing too efficiently in the model with comparison to the observations. With less (total) ice than C86, the ice particles are allowed to grow larger.

8) Page 19, line 7 – There may be fewer sources, but so little is known about INP in the Arctic that I think this sentence would be better removed.

- This sentence has been removed as requested.

9) Possibly of interest to the authors, INPs measured at Alert, Nunavut (Mason et al., Atmos. Chem. Phys., 16, 1637–1651, 2016) during spring to early summer vary from 0.2 per litre to 1 per litre for temperatures from 20oC to 25oC, which covers the range of average ice particle number concentrations you report for the three cases.

- We thank Reviewer 2 for bringing this study to our attention, we have included a reference to it in Sect. 5.4.3 (page 24, line 14 onwards):

  *" These concentrations compare well with INP measurements made at the Alert station in the Canadian Arctic in the spring of 2014, where mean INP number concentrations of 0.05 $L^{-1}$, 0.2 $L^{-1}$ and 1 $L^{-1}$, at -15 $^{o}C$, -20 $^{o}C$, and -25 $^{o}C$ respectively, were measured by Mason et al., 2016. "*

**Reviewer 3:**

The paper "Microphysical sensitivity of coupled springtime Arctic stratocumulus to modelled primary ice pack, marginal ice, and ocean" by Young et al. 2016 investigates the sensitivity of three Arctic mixed-phase clouds (at different surface conditions) to primary ice concentrations for three different types of primary ice parameterisations. The simulated microphysical properties of the three case-clouds are compared to field observations. The setup of the study is interesting and provides some nice insights about the modelling of Arctic mixed-phase clouds. However, some parts of the discussion are quite circuitous and could be a bit better organised to transport the main message of the paper in a more compromised way. In total the study is suitable to be published in ACP.

**General comments:**

- Some sentences of the abstract are not very clear for readers who haven't looked at the paper yet. More specifically in line 7 it is not very clear what kind of key sensitivities (of what to what) you are referring to. It also reads as if the key sensitivities emerge from the comparison to the observations, but they already emerge from the sensitivity simulations itself (or from a combination of both).

- We have updated the language to specifically refer to cloud ice number concentrations (page 1, lines 7-8):

   "*Three key dependencies on N_ice are identified from sensitivity simulations and comparisons with observations over the sea ice pack, marginal ice zone (MIZ), and ocean.*"

In line 11 it is not specified what kind of parameterisations you are talking about. You should add "for primary ice nucleation" (or something similar).

- We have included this information as requested (page 1, lines 10-12):

   "*We show that warm supercooled (-13 $^{\circ}$C) mixed-phase clouds over the MIZ are simulated to reasonable accuracy when using both the DeMott et al., 2010 and Cooper 1986 primary ice nucleation parameterisations.*"

In line 16 you could add half a sentence what "cloud break up" is.

- We have lengthened this sentence (page 1, line 17) to read:

   "*… promoting cloud break up through a depleted liquid phase*"

- The terming "primary ice crystals/primary ice number concentration" is not always consistent, e.g. on page 5, line 11 it is written "pristine ice crystals". It is also not clear how different the terming is meant in the different parameterisation schemes. In DeMott et al. 2010 the parameterisation scheme refers to INP, but INP translate directly to primary ice crystals (the way they mean it) and therefore the parameterisation schemes should technically be the same. If that is not the case, a better explanation (also of how it is differently implemented in the model) would be needed.

- We have used the terms as they appear in the parent articles to be distinct in their meaning, as the parameterisations predict subtly different things. Cooper 1986 was derived from measurements of ice number concentrations, whilst DeMott et al., 2010 was derived from INP measurements. The ice number concentration is the key quantity of importance to our study. For the analysis presented, we have chosen to refer to primary ice number concentrations. We felt it was important to highlight the fact that DeMott et al., 2010 was derived from INP measurements, and therefore should only be used to predict INP number concentrations. We are assuming that these all activate and, whilst this is potentially the

case, it is possible that the conditions attained were not suitable for all to do so. It is therefore an assumption that our D10 INP all activate. This is discussed in Sect. 5.4.

Following this comment, we decided to be more explicit in the manuscript when referring to ice number concentrations. N_ice is used to refer to ice number concentrations; for example, from evaluated parameterisations or observations. N_isg is used to refer to the total frozen (ice+snow+graupel) number concentrations simulated with the model. This distinction has been made to make it clearer what parameters are being compared at each step.

The terming "primary ice nucleation" does sound tautologous (nucleation always leads to primary ice).

- We agree that it is repetitive phrasing; however, it is often referred to in this manner and we wish to make the clear distinction that only primary processes are being considered.

- It would also be interesting to add a simulation where the different freezing pathways (immersion freezing and deposition nucleation) can compete and look at the importance of the different pathways acting in different S/T-regimes. I was missing this discussion or the discussion of this aspect in the manuscript. Of course it is interesting to look at the resulting ice crystal concentrations of both pathways separately but in reality both pathways could take place in parallel or rather compete with each other depending on the environmental conditions.

- The influence of deposition-condensation, immersion- and contact-freezing acting together is discussed in the Supplement, using Cooper 1986, Bigg 1953, and Meyers et al., 1992 respectively. When using these three parameterisations combined, the modelled microphysics is unrepresentative of the observations (Fig. S7). In case 1, no liquid is modelled when using all three parameterisations, and the ice phase agrees poorly with observations (similar to the C86-only cases presented in Sect. 4.1). Including Bigg immersion-freezing and Meyers et al., 1992 contact-freezing increases the ice number concentrations in case 2 to quantities much larger than observed ($\sim 2$ L$^{-1}$). In case 3, the inclusion of these immersion- and contact-freezing parameterisations causes a sharp increase in the modelled ice number concentrations; an increase which causes cloud glaciation. When immersion-freezing is represented by the Bigg parameterisation for freezing rain drops, the ice number concentrations are increased, thus producing poorer agreement with observations. A summary of these findings have been added to Sect. 4.1; however, the majority of the discussion has been kept in the Supplement.

We are not directly referring to immersion-freezing with our method, instead we are stating that water saturation must be attained before ice nucleation can occur. This simply means that we only allow ice crystals to form when there are also cloud droplets present. We do not specify a nucleation mechanism, instead we are assuming that any of condensation-, immersion-, or contact-freezing could be occurring under these conditions to give the resulting ice crystal number concentrations. If we were to assume that our limitation is representative of immersion-freezing alone, and used this additionally within the Morrison et al., 2005 microphysics scheme, we would simply get two sources of ice (from the deposition parameterisation and the immersion parameterisation) instead of just one. The Morrison et al., 2005 scheme therefore is not tailored to investigating the competition between different modes of freezing, as the represented modes simply add to the ice fields when these parameterisations are satisfied, independent of each other.

To investigate the competition between different freezing modes, the composition of aerosol would need to be explicitly represented to take into account the dominance of one freezing mode over another in different environmental conditions for different aerosol particle species. A spectral microphysics scheme may therefore be more appropriate to conduct this pathway of investigation; however, such schemes are rarely included in 3D models due to computational cost. To investigate the competition between the modes of freezing, we

suggest that a different scheme, which explicitly resolves aerosol composition and size distributions, would be necessary to provide an accurate representation of the competition between the microphysical processes occurring in these clouds.

- The result that ACC provides the best agreement with the observations (e.g. page 11, line 3) is trivial since the parameterisation is based on the observations. However, it is interesting to have an empirical parameterisation in the comparison, but maybe a bit more critical discussion on this could be added.

- We agree that the "performance" of the ACC relationship is poor phrasing, and have updated its use in the manuscript to be an "empirically-derived" comparison for the two established parameterisation. Reviewer 2 also highlighted this issue, and it has been addressed throughout the manuscript. We feel it is important to include the ACC results as it shows that persistent mixed-phase conditions can be attained with its use across all three cases. We address the limitations of this relationship in Sect. 5.5, but have included more critical discussion as requested. Specifically, we have focussed on the over predicted liquid phase and potential overestimation of cloud stability (i.e. would this relationship allow for cloud break up downstream?).

- The first part of the paper is quite lengthy, especially the comparison of every single case. Some plots do not seem very interesting at the first point but are very interesting later in the discussion when you explain some of the details behind some features (e.g. Fig. 8). I also had the feeling that later in the discussion many things are repeated (e.g. in section 5.4.3). The discussion itself was (just looking at the headers of the sections) not very intuitively organised, it is quite difficult to see where this now leads to/what the main points are/will be (before reaching the conclusions). It could help to restructure the paper/think again how to organise it so that the focus is clear and the paper interesting to read without losing interest in the first part.

- We had originally structured the article into distinct Results and Discussions sections to adhere to ACP's manuscript preparation policies. However, we agree that some restructuring would help to keep the reader motivated and emphasise the focus of the article. We have therefore restructured the Results and Discussions sections to make the focus of the article clearer to the reader throughout.

I had the feeling that most of the description on page 22 and also section 5.4.3 could already be moved to the case description and make this part more interesting to read. Also the comparison of the cases within each other came quite late (before they seemed to be quite isolated in the analysis). However, this point might be subjective and a matter of taste.

- Comparisons between simulations that were detailed in Sect. 5.4.3 have now been moved to the relevant case results section to make the focus clearer.

**Specific comments:**

- Page 2, line 15: Add which kind of parameterisations you are talking about.

- Sentence has been updated to refer specifically to ice nucleation:

"*...commonly-used mid-latitude parameterisations for primary ice formation, such as...*"

- Page 2, line 20: You should add in brackets the name of the four ice nucleation modes you talk about. You could also add a bit more explanation (or more structured explanation) about the different ice nucleation modes. You mention that partly later, but maybe it would help the inexperienced reader if you have short explanations first before you elaborate the pathways and their representation in models in detail.

- This paragraph has been expanded to reflect Reviewer 3's comments. A sentence giving an approximate description of each nucleation mode has been included. This paragraph has also been reorganised to accommodate these changes (page 2, lines 22 - 30).

- Page 2, line 20: In most models not all four ice nucleation modes are represented. It is a bit miss leading to say "commonly represented". Many models only treat immersion freezing, on the contrary contact freezing is very rarely explicitly modelled.

- This section has been reworded to describe the modes physically instead of in a modelling context (page 2, lines 22-25).

  *"Primary ice particles may be nucleated heterogeneously through four different modes of action; deposition, condensation, immersion, and contact (Pruppacher and Klett, 1997). These modes describe the deposition of water vapour onto an INP, forming ice directly (deposition) or freezing upon condensation (condensation), or the freezing of a cloud droplet through activation from within (immersion) or collision with an INP (contact)."*

- Page 2, line 26: Your example does not fit to the argumentation before (referring to deposition nucleation instead of immersion freezing).

- The argument was correct, but we agree that not enough information was included. By suppressing the deposition mode, we were implying that the immersion mode would likely occur. In our reorganisation of the manuscript, we decided to remove this discussion from the manuscript as the introduction had become quite lengthy and this information was not strictly necessary to our study.

- Page 2, line 28: You write that deposition nucleation and condensation freezing are experimentally difficult to distinguish but most instrument have rather difficulties to distinguish immersion and condensation freezing since in both cases the liquid phase is involved. Deposition nucleation takes place at a different saturation ratio/temperature regime compared to condensation freezing. Many models use immersion freezing as a surrogate for immersion and condensation freezing. I was surprised to read that deposition nucleation should be often related to condensation-freezing- are you referring to pore-condensation? Maybe this issue is related to the definition used of the nucleation modes and more a matter of phrasing/language but it could be confusing to other readers as well, so it might be better to add more explanation.

- We were referring to measurements made in the past through which parameterisations have been derived (e.g. Cooper 1986, Meyers et al., 1992). These articles refer to deposition-condensation nucleation as they could not guarantee one mode was occurring without the other. We agree that the immersion-condensation freezing argument is similar, especially with the most up-to-date INP counters which utilise water saturated conditions within their chambers. We are not specifying which nucleation mode we are referring to as we are simply making ice nucleation occur at water saturation; in reality, this could encompass immersion-, condensation-, and/or contact-freezing. For the purpose of this article, we simply wish to acknowledge the difficulties in measuring specific ice nucleation modes and knowing how others have done it in the past. We agree that in its current state this argument could be confusing; therefore, we have adapted the language to make this distinction clearer (page 2, lines 26-28).

  "*Due to their similarities, it can be difficult to differentiate between these mechanisms in measurements; for example, deposition or immersion nucleation are often quoted to occur alongside condensation-freezing processes (e.g. Cooper 1986; Meyers et al., 1992; de Boer et al., 2010; Fan et al., 2016)."*

- Page 2, line 10: What do you mean when you say "ice number concentrations will be suppressed under these conditions"? So deposition nucleation is also only allowed to take place at water saturation? That does not make sense physically.

- As above, we are not specifying a nucleation mode. We are using the location of the deposition-condensation parameterisation in the microphysics scheme to input an ice nucleation parameterisation; this parameterisation is not always used under deposition conditions, in fact we restrict the parameterisation to water-saturated conditions for the majority of the model runs presented. We used this language so that those familiar with the scheme could follow, but we agree that this may have been misleading. We have updated the language (page 3, lines 12-14) to make this clearer.

  *"We hypothesise that ice number concentrations will be suppressed and liquid fractions will be enhanced under this restriction, thus reducing the influence of the WBF mechanism and prolonging cloud lifetime."*

- Page 4, line 20: What is a sub-Arctic McClatchy profile? Either explain or generalise?

- A McClattchey dataset includes historical measurements of stratospheric transmittance. These profiles allow the optical properties of the simulated atmosphere to be representative of the region modelled, and are used to determine the attenuation of radiation by the atmosphere. Vertical profiles of tropospheric temperature, pressure, water vapour, and ozone were imposed. The sub-Arctic versions used are relevant to this region alone and differ, for example, from similar measurements in the tropics. Additionally, the spelling of McClattchey has been rectified (page 4, lines 26-28).

  *"Sub-Arctic McClattchey profiles of tropospheric temperature, pressure, water vapour, and ozone -- based on historic measurements of stratospheric transmittance -- were imposed in all simulations to ensure the initialised vertical profiles were representative of the environment modelled."*

- Page 4, section 2.2: What is the time step of the simulations? 150 seconds?

- The time step is variable to satisfy CFL criteria and varies between case studies. For cases 1, 2, and 3, dt was approximately 0.3 sec, 0.4 sec, and 0.2 sec respectively. This information has been included in the manuscript (page 4, line 25-26).

- Page 5, line 3-7: You can skip this explanation since you switch of Bigg 1953 and Meyers 1992 in the final simulations. That is confusing (especially when readers watch out for B53 and M92).

- We wish to keep this information in, again, for those familiar with the microphysics scheme. It is important to the study to know that these have been specifically switched off. We have removed the short-hand for the parameterisations and reworded this paragraph to reflect this comment. We have moved this discussion to Sect. 4.1 (page 10, lines 3-17).

- Page 6, line 11: It is good that you point out the limitations of this study. However, it would also be nice to add how realistic this idealistic study is and under which conditions you would have a similar system in reality.

- The model setup is idealistic in terms of the surface conditions and the representation of ice nucleation; therefore, we wished to stress this to the reader. However, the model is able to produce cloud microphysics comparable to the observations in some cases, simulating clouds which persist for lengths suggested by previous observations. This suggests that our assumptions may be credible, creating an environment that is more representative of reality than first thought.

Whilst INP depletion may be more applicable to clean environments susceptible to INP plumes, INP recycling below cloud and/or a source of INP at the surface/aloft could act to continually supply INP to the cloud via advection or entrainment. This study could act to represent such an environment with a consistent source of INP.

We have added information to this effect into the manuscript (page 6, lines 12-14):

"*However, this setup can give an approximation of the cloud microphysics that may form in the vicinity of an INP source; for example, a local source at the surface or a long-range transported INP population aloft.*"

- Why did you only add an ice crystal number concentration sensitivity for DeMott et al. 2010? Is there a reason you picked this parameterisation and did not add it for all of them (computational costs?)?

- Yes this was predominantly due to computational cost and time constraints. We wished to include sensitivity tests, and decided to choose the most recent parameterisation for these, which is based on measurements using updated techniques. This has been added into the manuscript for clarification (page 6, lines 19-20):

  "*We chose D10 for this sensitivity study as this is the more up-to-date of the two established parameterisations used.*"

- Fig. 1: You should plot the parameterisation only in the temperature range where they are valid or make them transparent in the temperature regime beyond their validity. Or do you extrapolate the parameterisations schemes over the whole temperature range in your model setup (then I miss interpreted it wrong before)?

- We have updated Fig. 1 to show the parameterisations in their valid temperature ranges only. Reviewer 3 is correct, these are not extrapolated outwith this range (except for ACC in case 3, this is discussed in Sect. 5.5 as being a point to note about this simulation).

- Fig. 1: Instead of having three line for D10 and the corresponding variations, you could only plot D10 and add a shaded area around the line. The D10 Fit is not needed. You also do not really discuss it in detail later. The D10x0.5 line is not really needed here, you already describe it later (and no visualisation is necessary). However, of course you can keep it, it would just make the figure a bit less busy.

- Figure 1 has been updated following Reviewer 3's comments. We were unsure of what shaded area was requested; therefore, we have opted to show the variability (due to different aerosol loadings) in each version of D10 (D10x0.1, D10, and D10x10).

- Fig. 2: You could increase the figure to enable better readability. Remove the doubled red altitude axis on the right (or colour it black), that is miss leading. It would be useful to add the cloud extent in the figure. Would you still need the grey boxes when you add the cloud extent to the figure (since the altitudes without sampling seem to be always below cloud)? Would it be possible to use the same scale for all cases?

- Reviewer 3's comments did not apply to Fig. 2, so we assumed (through their comments) that they were referring to Fig. 4. We would like the current grey boxes to remain as, although these altitudes do appear to be below cloud across the three cases, we cannot say for certain what was present at these altitudes because we didn't sample there. We would like to illustrate that surface fog or a low-altitude cloud layer, for example, cannot be ruled out. We have removed the red axis and have altered the scales as requested. Cloud extent is inferred by the measurements, as only in-cloud measurements are shown.
* * *
- Page 10, line 14 and line 17: At which altitudes are the ice crystal concentrations estimated?

- For case 2, the concentrations are taken from approximately 1000m, at the end of the simulation. For case 3, these are taken from ~1450m. This information has been added to the manuscript (page 10, lines 27-28, lines 29-30).

   *"Modelled N_isg over the MIZ (~1.0 L$^{-1}$ at 1000 m, Fig. 5b) is in reasonable agreement with the mean observed (0.35 ± 0.20 L$^{-1}$, Table 2)."*

   *"Such conditions are also attained in case 3 (Fig. 5c); modelled N_isg peaks at 3.7 L$^{-1}$ at ~1450 m, whereas only 0.55 ± 0.95 L$^{-1}$ was observed."*

- Fig. 5: You could increase the figure to enable better readability. The scale is not reasonable (there should not be negative Q_liq).

- Figure size and scale amended as requested.

- Page 12, line 6: Can you further explain this?

- The shaded areas in Fig. 7 represent the variability in the Q_liq profile over the given time window. In contrast to the N_ice>100um data, clear contours of shading are not present around the plotted profile. This is what we were alluding to with our analysis; however, we felt this comment is not necessary (and confusing) and so have removed it from the manuscript

- Page 12, line 8/Figure 8: Again is the D10 Fit really needed here?

- The D10 simulation shown in Fig. 8 is the D10 run for each case, not the fit from DeMott et al., 2010 that was shown in Fig. 1. Figure 8 has now been dissected to show each parameterisation for each case (Figs. 6 and 8 now make up Figs. 6, 9, and 10); therefore, this query should no longer be an issue.

- Page 12, line 14: Why does IWP decrease subsequently?

- The IWP decreases once the ice number concentration begins to be depleted in the cloud layer, through fallout as snow (page 13, lines 11-12).

   *"… the simulated IWP increases initially (between approximately 17 h and 20 h), but subsequently decreases as the N_isg falls out from the cloud layer."*

- By how much is N_{ice > 100 mum} still clearly related to the ice nucleation parameterisations in your model? Which other processes might influence this variable? Is it fair to compare this variable among the different parameterisations (since that is not the size of primary ice formation)?

- N_{ice > 100 μm} is the number concentration of ice crystals larger than 100um in size only. We believe this is a robust parameter to compare with observations as we can be confident that we are comparing ice crystals over the same size range. As for the comparisons between simulations, the only differences are the parameterisations themselves; therefore, any additional processes which act in one simulation will act in all of the simulations. We feel that N_{ice > 100 μm} is the best parameter to compare with the observations, whilst N_isg (total ice+snow+graupel number concentration) is more suited for comparisons between the model simulations. We have altered the language throughout the manuscript to make this distinction clearer.

- Fig. 6: Would it be possible to use the same scale for all cases?
* * *
- Figure 6 has been dissected into each separate case following comments from the other reviewers. In each separate case, the 5 simulations (for each parameterisation) are shown on the same colour scale.

- Fig. 7: It is very difficult to compare the single lines. It would help to increase the size of the figure. It would also be possible to cut Fig. 7 a, b, d, e, g, h to the relevant altitudes (leaving away everything above 1000 m). It is unclear over which time span the mean was taken and why. What was the availability of the observations? Did you choose to calculate the mean as described to temporally collocate the data?

- Following Reviewer 2's comments, the first row (7a-c) has been removed, making the 6 remaining sub-figures clearer. For the observations, the mean is taken over all measurements of the relevant cloud layer. The mean at each altitude bin was taken over approximately one full hour of data; however, each altitude was sampled differently due to constraints on the flight track (i.e. vertical profiles were not taken, a selection of straight-and-level runs and sawtooth profiles makes up the data from which the mean is calculated). This observed mean (at each altitude) was then subtracted from the modelled mean (per hour) for the full 24h simulation to find which time step gave the best agreement with the observations, allowing for the best possible comparison for each case. The time step selection is detailed in the supplement.

- Page 15, line 3: Why do the glaciation events take place every 3h? What is driving that?

- High concentrations of ice crystals are produced which use up the water vapour in the air through depositional growth, causing the ice supersaturation to decrease. Therefore, large number concentrations of ice crystals are produced when ice supersaturation is reached, these deplete the vapour, fall out of the cloud and sublimate, and the ice crystals form again when ice supersaturation is reached (which occurs readily due to the cold temperatures modelled). This explanation has been included in the manuscript for clarity (page 17, lines 20-23).

  *"Due to the strong dependence of $N$ ice on temperature, high $N_{isg}$ are created which readily undergo depositional growth, deplete the vapour field, and fall from the cloud once the particles transition to the snow category. The vapour field recovers due to the moisture fluxes from the surface, and the process repeats once water and ice supersaturation are attained."*

- Page 15, line 13 + page 15, line 17: You do not really mention or explain the spikes here which is a bit irritating. You also do not explain (here) why only D10 has these glaciation peaks. It is also unclear (here) why in case of C86 the glaciation leads to a decrease of IWP and not an increase. You could think of reorganising your paper so that you add already part of the discussion here.

- As above, these sections have been reorganised to make these points clearer to the reader.

- Page 15, line 22: Explain what W is.

- W represents the vertical velocity. This has been updated in the manuscript (page 18, lines 11-12):

  *"… and the vertical velocity, W, is chosen at approximately cloud top (1500 m)."*

- Fig. 8: Was more interesting later for the discussion but at this part of the paper it does not seem so interesting, you might want to shift either the figure or the discussion (see general remarks). Fig. 9: The features (peaks) are not really clear until the discussion. You could add the periods at the time scale when the cloud had a mixed-phase structure/when there was a cloud. Fig. 10: The

—————————————————————————————————————————————

differences of the last three lines did not get very clear before the discussion also includes the precipitation (Fig. 11)- it could help to reorganise the discussion here.

- As above, we have restructured the article to include some discussion in the same section as the figure to motivate the reader.

- Page 15, line 33: The unit for the precipitation is a bit confusing here (is clear in Fig. 11). Fig. 11: It would be an interesting information to also add the total amount of precipitation for all the cases (mm/mˆ2).

- This may be misleading, as the rain produced is effectively virga as it evaporates below cloud and does not reach the surface. We consider large particles, which are falling relative to the cloud layer, to be precipitation, and it is the number concentration of these particles that is of interest to our analysis. We agree that using the term "precipitation" here may be misleading; therefore, we have adapted our language throughout to refer to "precipitable particles" and "large hydrometeors".

- Page 21, line 2: You could elaborate here why it is so different in case 3. What is different in that case?

- The dropsonde data used to initialise the model gives a moist boundary layer in which cloud forms immediately. This cloud is unsustainable due to its mixed-phase, high N_ice nature, and it begins to decay by the WBF mechanism (added to page 16, lines 1-2).

  *"Cloud forms and begins to decay immediately in case 3, as shown by the decreasing LWPs modelled (Fig. 8c), caused by the moist BL and a high N_isg which acts as an efficient sink for liquid by the WBF mechanism."*

- Page 21, line 14: It is good that the modelled N_ice is in reasonable agreement. However, if that is due to colder temperatures than observed, it would indirectly mean that the temperature dependence of the parameterisation schemes used (or the temperature regime where they are efficient) is not correct. You could add some critical thoughts about this issue.

- As Fig. 7 has now changed, we have rephrased this argument to avoid confusion. Whilst the absolute number concentration of total ice particles is in reasonable agreement with observations (Tables 2, 3), it is important to note that the N_{ice>100 µm} agreement is poor (Fig. 7b). We have addressed this issue in the manuscript, and added some critical discussion as requested (page 22, lines 9-13):

  *"As a result, the N ice>100µm modelled with the temperature-dependent parameterisations considered is greater than observed (Fig. 7b). Overall, the N_isg is in reasonable agreement with the observed N ice (Tables 2, 3), likely due to the low concentrations of snow and graupel produced at the warm sub-zero temperatures considered, and it is probable that this agreement would improve further if the modelled temperature was accurate. In contrast to cases 1 and 3, the reasonable agreement of N_isg and poorer agreement of N_{ice>100µm} suggests that the ice crystal growth rates are too efficient in case 2."*

- Page 23, line 12: I did not understand what you meant by "sweet spot" here when you mentioned it the first time. However, it was clear later on.

- This reference has been updated to be clearer (page 21, lines 12-13):

  *"...there is an optimal N_ice for cloud persistence in this case…"*.

—————————————————————————————————————————————

The "sweet spot" reference in the Conclusions has been kept the same to emphasise its importance.

- Page 25, line 5: Add the order of magnitude.

- We have restructured this paragraph and the comment in question has been removed as a result. We have ensured a range for ice number is quoted in the revised paragraph. This discussion is now Sect. 5.6, on page 24.

- Page 26, line 2-4: Could the less pronounced difference in case 2 (compared to 1 and 3) be a result of the higher cloud top temperature and the onset temperature of freezing? It would be an interesting aspect to add to the discussion.

- Yes, with a higher CTT, less ice will form in the cloud in the absence of secondary ice production. We have included this aspect in our discussion in Sects. 5.3 and 5.7.

**Technical corrections:**

- Page 1, line 12: The Cooper (1986) parameterisation…

- Changed as requested.

- The ice nucleation pathway deposition nucleation is commonly not named deposition freezing, since it does not involve the liquid phase (and freezing refers to the liquid phase).

- Changed throughout the manuscript as requested.

- Page 3, line 14: Replace guide by guidance.

- Changed as requested (now page 3, line 18).

- Page 3, line 32: You could exchange the second with the first sentence.

- Changed as requested (now page 4, lines 3-5).

- Page 4, line 18: The numbering of the cases is wrong in this case (here it reads as if the ocean case is number 2 and the marginal ice zone case is number 3). You should check if the case-numbering is consistent everywhere.

- Thank you for highlighting this mistake, we had missed it. Case 2 is indeed the MIZ case, and case 3 is over the ocean. We have checked the rest of the document and all other references are correct.

- Page 5, line 2: You write that you vary the form of the deposition-condensation freezing parameterisation but you only use C86 and compare it to D10, which refers to immersion freezing. You should therefore change this here to prevent confusion.

- We were referring to the location of the changed relationship in the Morrison microphysics scheme for those who are familiar with it. As stated, primary ice nucleation is included as three separate parameterisations in this scheme: we removed the separate immersion- and contact-freezing relationships so that the ice could only be formed by the parameterisation that we chose. This change occurred in the position of the deposition-condensation nucleation parameterisation in the code. We see how this may be misleading, however, so we have made this more explicit in the manuscript. This discussion has been moved to Sect. 4.1 (page 10, lines 7 - 17).
* * *
- The equations have the unit and the dependence in the same bracket, which is a bit strange. It would be more correct to write the unit and the dependence in separate bracket, e.g. N_ice(T_k) [mˆ{-3}].

- Equations 1-3 have been updated with Reviewer 3's suggestions.

- Page 5, line 19: It would read better if you have the text first and then the formula.

- This section has been re-ordered as requested.

- What does the Index k means for the temperature? Why do you not write T?

- The subscript K refers simply to Kelvin. Given that temperature is referred to in both Kelvin and Degrees in the article, we felt it necessary to specify which is being used in the evaluation of each relationship. This has been made clearer in the manuscript.

- Page 7, line 5: Replace These by The (otherwise the reference is missing).

- Changed as requested (now page 8, line 4).

- Page 7, line 12: Skip "over the".

- Changed as requested (now page 8, lines 18-19).

- Page 9, caption Fig. 4: Replace mixed-phase cloud by mixed-phase clouds.

- Changed as requested.

- The numbering/organisation of the figures is not always consistent (e.g. Fig. 7 does not use i/ii for the columns). It would be great to have the explanation of the numbering of the different columns as in the caption of Fig. 10 in the caption of Fig. 1 (you do not need it then in Fig. 10). In Fig. 12 you use the labels i/ii in a different way then before.

- Line plot subpanels are labelled a-f. Colour plot subpanels have an additional label of (1-4) for different data presented from the same simulation. These colour figures are labelled in this way to make it clear that these subpanels are related, e.g. a(1) and a(2) are related and from the same simulation. Figure labels have been updated to follow these guidelines. We have also shortened the figure captions of later figures as requested, by referencing earlier figures.

- Page 20, line 11: Replace , before Mason by ;.

- Changed as requested (now page 23, line 25).

- Page 20, line 30: More accurately would be mixed-phase formation phase instead of formation phase.

- We agree with this requested change; however, on reordering the manuscript, we have removed this reference as it was not necessary to our arguments.

- Page 24, line 30: Move also in between are and not.

- This statement has been moved to Sect. 4.2.3, and re-worded so that the requested change is no longer necessary (page 15, lines 10-13):

"*ACC produces comparable $N_{ice>100\mu m}$ and $Q_{liq}$ to observations as expected – when not considering the shattering event at cloud base (Fig. 4c) – and predicts 0.54 $L^{-1}$ at the*
* * *
*case 3 CTT. D10×0.1 produces reasonable agreement with the Q_liq observations at 7 h (Fig. S11); however, the rapidly increasing cloud top height and Q_liq with time are not representative of the observations.*"

- Page 26, line 7: Replace microphysical structure by microphysical structure of MPS.

- Changed as requested (now page 25, line 26).

[revised manuscript text omitted]

considered. To force the formation of persistent liquid in all cases, we restrict the formation of primary ice to water-saturated conditions in our subsequent simulations.

**4.2 Ice nucleation at water-saturation**

**4.2.1 Case 1: Sea ice**

5   Figure 6 shows modelled $N_{\overline{ice}\ \underset{\sim}{isg}}$ and liquid water mixing ratio, $Q_{liq}$, using the  five parameterisations – D10×10, C86, D10, ACC, and D10×0.1 – over the sea ice. Vertical (Z-Y) slices of $N_{\overline{ice}\underset{\sim}{isg}}$, $Q_{liq}$, and W at 21 h are included in the Supplement (Fig. S8).

    No liquid water is simulated when using D10×10. A mixed-phase cloud is simulated ~~at ~500~~below 600 m after 17 h in the remaining four simulations, with a liquid layer at cloud top with ice formation and precipitation below. Peak $Q_{liq}$ varies

10   from C86 at the smallest (0.09 g kg$^{-1}$), through D10 (0.1 g kg$^{-1}$)  and ACC (0.14 g kg$^{-1}$), to D10×0.1 at the largest (0.16 g kg$^{-1}$, Table 3). $N_{\overline{ice}\ \underset{\sim}{isg}}$ and $Q_{liq}$, with the exception of D10×10, both increase with time as each cloud evolves. Modelled $N_{\overline{ice}\ \text{is of the same}\ \underset{\sim}{isg}\ \text{varies through an}}$ order of magnitude, with maximum values of

[Figure]

**Figure 6.** Simulated total ice number concentrations ($N_{isg}$, **1**) and liquid water mixing ratios ($Q_{liq}$, **2**) using the (**a**) D10×10, (**b**) C86, (**c**) D10, (**d**) ACC, and (**e**) D10×0.1 parameterisations for case 1 (sea ice). All are restricted to water-saturation. Run length 24 hours. Temperature (°C) contours are overlaid in white. Runs are arranged such that the simulation which produced the most ice (D10×10, **a**) is on the top row, and that which produced the least ice (D10×0.1, **e**) is on the bottom row. Note changing colour bar at the top of each column, which corresponds to data in that column only.

2.89 L$^{-1}$, 2.32 L$^{-1}$, 1.29 L$^{-1}$,  0.47 L$^{-1}$ , and 0.13 L$^{-1}$ attained by D10×10, C86, D10, ACC, and D10×0.1 respectively.

Figure 7 shows a comparison between measured and modelled $_{ice}$, N$_N$$_{ice>100\mu m}$  and $Q_{liq}$ for each case when using

**Table 3.** Maximum modelled values during each case for each parameterisation implemented at water-saturation.

| Case | Parameter | D10×10 | C86 | D10 | ACC | D10× 0.1 |
|---|---|---|---|---|---|---|
| Sea ice (case 1) | $N_{\text{ isg}}$ [L$^{-1}$] | 2.89 | 2.32 | 1.29 | 0.47 |  0.13 |
| | $Q_{\text{liq}}$ [g kg$^{-1}$] | 0 | 0.09 | 0.10 | 0.14 |  0.16 |
| MIZ (case 2) | $N_{\text{ isg}}$ [L$^{-1}$] | 6.57 | 1.09 | 1.03 | 0.36 |  0.11 |
| | $Q_{\text{liq}}$ [g kg$^{-1}$] | 0.12 | 0.29 | 0.28 | 0.34 |  0.39 |
| Ocean (case 3) | $N_{\text{ isg}}$ [L$^{-1}$] | 15.5 | 3.83 | 3.01 | 0.71 |  0.37 |
| | $Q_{\text{liq}}$ [g kg$^{-1}$] | 0.10 | 0.32 | 0.32 | 0.36 |  0.38 |

 produces the greatest ice number concentration, with  D10  producing the least (Fig. 7a). ACC provides the best agreement with the mean observed $N_{\text{ice}}$, simulating approximately 0.4 L$^{-1}$, and ACC.

5   2DS data  has poor resolution at small sizes (<80 μm), preventing the particle shape factor from being accurately determined at these sizes (Crosier et al., 2011; Taylor et al., 2016; Young et al., 20 therefore, the number concentration of small ice crystals is not a reliable measure with this instrument. For this reason, the observed number concentration of ice crystals greater than 100 μm are  directly compared with modelled ice and snow particles in this size range. Figure 7a shows this comparison using the C86, D10, and ACC

10  parameterisations for case 1. Mean parameters modelled at 21  h during case 1 are shown in Fig.7(a, d). The empirically-derived ACC relationship produces $N_{\text{ice>100μm}}$  and $Q_{\text{liq}}$  profiles comparable to the mean observed as expected (Fig. 7a, d),

15  suggesting that ice particle growth rates are adequately represented, whilst D10

 and C86 overpredict $N_{\text{ice>100μm}}$ and marginally underpredict $Q_{\text{liq}}$. Comparisons including D10×10 and D10×0.1

20   and the method for choosing these time steps are detailed in the Supplement (Figs. S11, S12).

Liquid and ice water paths (LWP and IWP, respectively) using each parameterisation are shown in Fig. 8(a, d). Both increase with model time when using each of the parameterisations. D10×0.1 produces the highest LWP and lowest IWP. D10×10

25  produces no liquid – giving a LWP of zero – and the simulated IWP increases initially (between approximately 17 h and 20 h),

[Figure]

**Figure 7.** Observed $N_{ice>100\mu m}$ and $Q_{liq}$ for the sea ice (**column 1**), MIZ (**column 2**), and ocean (**column 3**) cases. Observations are shown as black boxes, similar to Fig. 4. Mean modelled concentrations of ice and snow particles greater than $100\,\mu m$, using the C86 (magenta), D10 (green), and ACC (blue) parameterisations, are overlaid. Model time steps of $21\,h$, $17\,h$, and $7\,h$ are used for the sea ice, MIZ, and ocean cases respectively, as these time steps offer the best comparison with the observations. Shading (in pink, green, or blue for C86, D10, and ACC respectively) indicates variability in the model parameters from $\pm 3\,h$ in cases 1 and 2, and $\pm 4\,h$ in case 3, where a larger interval is implemented in the latter case as the chosen parameters showed little variability over the shorter time step. In panel (**f**), the variability illustrated is always less than the mean modelled profile shown using each parameterisation as the $Q_{liq}$ is at its greatest at the chosen time step. Observed $N_{ice>100\mu m}$ data from noted shattering event (Young et al., 2016a) are excluded in panel **c**, so that only primary contributions of ice are considered.

but subsequently decreases as the $N_{isg}$ falls out from the cloud layer. The D10 and C86 parameterisations produce similar trends in the LWP and IWP traces, resulting in approximately $15\text{-}20\,g\,m^{-2}$ and $2\text{-}3\,g\,m^{-2}$ respectively by $24\,h$.

Negligible surface fluxes were applied in this case; therefore, cloud dynamics was driven primarily by longwave radiative cooling (similar to Ovchinnikov et al., 2011). In the observations, a lack of strong turbulent motions within this mixed-phase cloud layer caused a suppressed LWMR in the vicinity of moderate ice number concentrations (Young et al., 2016a). The LEM reproduces these conditions well in the absence of strong surface fluxes, as a small $Q_{liq}$ and a reasonable $N_{isg}$ are modelled under the restriction of water-saturated ice nucleation.

**4.2.2 Case 2: Marginal ice zone**

[Figure]

**Figure 8.** Vertically-integrated liquid (**a-c**) and ice water paths (**d-f**) for the sea ice, MIZ, and ocean cases when implementing each of the C86, ACC, D10, D10×10, and D10×0.1 parameterisations under water-saturated conditions.

All parameterisations produce a mixed-phase, sustained cloud layer over the MIZ (case 2, Fig. 9). Modelled LWPs and IWPs are larger in case 2 than in case 1. Strong surface fluxes are implemented in case 2 to represent a comparatively-warm ocean at the surface, allowing turbulent motions to sustain a greater $Q_{liq}$ within the mixed-phase cloud layer (Morrison et al., 2008).

Figure 9 shows that there is little variation between the simulations  except when implementing
5  D10×10. $N_{isg}$ of up to 6.6 L$^{-1}$ are simulated using D10×10, with a suppressed $Q_{liq}$ (Fig. 9a). C86 and D10  perform similarly, predicting a $N_{ice}$ of 0.23 L$^{-1}$/0.34 L$^{-1}$ respectively at the CTT (Table 1), and producing comparable peak $N_{ice~isg}$ and $Q_{liq}$ values (Table 3)  when implemented in the model. Similar liquid (∼100 g m$^{-2}$) and  ice water paths (∼7 
[revised manuscript text omitted]

**Supplementary Material**

**Dropsonde dry bias**

A possible dry bias was influencing the dropsonde data used to initialise the LEM, producing a drier boundary layer than was observed. Of the data used to initialise the model, only the $q_{vap}$ field (as shown in Fig. 3) was affected. Cloud fields
5  are initialised with an adiabatic liquid water mixing ratio profile; therefore, this bias only has a small effect on the modelled cloud structure. However, the rate at which precipitation develops is affected and highlights an additional sensitivity of cloud structure to the humidity of the boundary layer in each case.

[Figure]

**Figure S1.** Relative humidity (RH) dropsonde measurements. Original data (black) was used to initialise the LEM in the presented simulations. Revised data is shown in red.

    Figures illustrating the extent of this dry bias are included as follows. Relative humidity data are shown in Fig. S1. An equivalent version of Fig. 3 is included to demonstrate the revised $q_{vap}$ profiles (Fig. S2). Additionally, an example test
10  simulation with revised dropsonde profiles from each case is included for justification.

[Figure]

**Figure S2.** As Fig. 3 with revised $q_{vap}$ data shown by a dashed (red) line in panels a, c, and e.

Figure S3 shows the equivalent simulation to Fig. 5(**a**) with the revised humidity profiles. Our conclusions remain unchanged with this new data: using C86 under the deposition-condensation conditions commonly used in WRF causes the production of an ice cloud with complete suppression of the liquid phase. The mixed-phase conditions observed are inadequately reproduced using these criteria.

[Figure]

**Figure S3.** As Fig. 5(**a**) with revised dropsonde initialisation conditions.

5      Figure S4 illustrates that increasing the humidity of the boundary layer quickens the formation of the mixed-phase cloud layer in case 2. Cloud top increases to higher altitudes than in the drier case (Fig. 9(**c**)), and reaches colder temperatures, due to the development of precipitation. The increased humidity allows for more efficient precipitation development, which acts to deplete the liquid phase of the cloud.

[Figure]

**Figure S4.** As Fig. 9**(c)** with revised dropsonde initialisation conditions.

Figure S5 shows the D10 ocean simulation (as Fig. 10**(c)**) with the revised initialisation profiles. In contrast to the drier conditions, this cloud glaciates by 22 h and all ice falls out of the cloud by 24 h. No convective features form as the precipitation which forms does so more efficiently with increased humidity, leading to complete cloud break up. Greater number concentrations of large solid hydrometeors (snow and graupel) are simulated at earlier times, leading to cloud break up with no convection
5    development.

[Figure]

**Figure S5.** As Fig. 10**(c)** with revised dropsonde initialisation conditions.

[Figure]

**Figure S6.** As Fig. 12(a) with revised dropsonde initialisation conditions.

**Bigg immersion-freezing and Meyers contact-freezing**

The influence of immersion- and contact-freezing within the Morrison et al. (2005) microphysics scheme was tested to quantify their contribution to $N_{\overline{ice}_{isg}}$. Simulations with contact-freezing (Meyers et al. 1992 - hereafter, M92) and immersion-freezing (Bigg 1953 - hereafter, B53) switched either on or off are shown in Fig. S7. The addition of B53 and M92 produces a signif-
5 icantly larger ice crystal number concentration (up to $3\,L^{-1}$, $1.5\,L^{-1}$, and $10\,L^{-1}$ in cases 1, 2, and 3 respectively) than the mean observed ($0.47 \pm 0.86\,L^{-1}$, $0.35 \pm 0.20\,L^{-1}$, and $0.55 \pm 0.95\,L^{-1}$ respectively, Table 2).

Modelled ice number concentrations with and without B53 and M92 active are similar in case 1. Both representations cause glaciation, and liquid water is not modelled at any point during the simulations. No improvement can be seen in the liquid water mixing ratio when both the B53 and M92 nucleation mechanisms are disabled. Modelled ice number concentrations for
10 case 2 peak at $\sim$1.5 $L^{-1}$ and $\sim$0.8 $L^{-1}$ with and without both B53 immersion- and M92 contact-freezing nucleation active. Both scenarios allow for liquid water to form in the cloud, with $\sim$0.2 g $kg^{-1}$ modelled. When B53 and M92 are active in case 3, high ice number concentrations are rapidly simulated at approximately 12 h-14 h. This event causes the evaporation of all simulated liquid water, and the region of high ice number concentration dissipates back to the original sustained concentration of $\sim$2 $L^{-1}$ afterwards. This event is not simulated when B53 and M92 are disabled, suggesting these additional sources of ice
15 number are the cause of this phenomenon.

[Figure]

**Figure S7.** Simulated ice number concentrations ($N_{\overline{ice}\,isg}$, 1) and liquid water mixing ratios ($Q_{liq}$, 2) using the Cooper (1986) parameterisation under default WRF conditions (T < -8°C, $S_w$ > 0.999 or $S_i$ > 1.08).  B53, M92, and C86 active.  C86 deposition-condensation  nucleation only.  Anomaly between simulations including B53 and M92 and those using C86 only.  Sea ice (case 1),  MIZ (case 2),  Ocean (case 3). Run length 24 hours. Temperature (°C) contours are overlaid in white. Note changing colour bars for each subfigure.

**Supplementary figures**

[Figure]

**Figure S8.** Z-Y slice of modelled $Q_{liq}$ (**top row**), $N_{isg}$ (**middle row**), and vertical velocity (**bottom row**) at 21 h over the sea ice (case 1). The $N_{isg}$ and $Q_{liq}$ fields are homogeneous, with liquid layer at cloud top and ice formation throughout. Enhanced turbulent activity, due to the comparatively larger liquid water content, is modelled with ACC (panel **c**). Note changing colour bars for each subfigure.

[Figure]

**Figure S9.** Z-Y slice of modelled $Q_{liq}$ (**top row**), $N_{\overline{ice\ isg}}$ (**middle row**), and vertical velocity (**bottom row**) at 21 h over the MIZ (case 2). Significant turbulence is simulated within the cloudy layer (bottom row). With comparison to the sea ice case, the liquid layer at cloud top is more heterogeneous in all cases. This is particularly clear in the D10 simulations (panel **b**), where $N_{\overline{ice\ isg}}$ is enhanced in downdraughts. Note changing colour bars for each subfigure.

[Figure]

**Figure S10.** Z-Y slice of modelled $Q_{liq}$ (**top row**), $N_{\overline{ice}\,isg}$ (**middle row**), and vertical velocity (**bottom row**) at 21 h over the ocean (case 3). Large updraught columns are simulated using D10, which correspond spatially with columns of high $Q_{liq}$. These updraughts are co-located with a precipitating (snow) region, evident from the $N_{\overline{ice}\,isg}$ figures (second row). C86 (panel **a)** had dissipated by 21 h; therefore, little activity can be seen in this simulation. Similar to cases 1 and 2, ACC produces a homogeneous liquid layer at cloud top, with ice below (panel **c**). Note changing colour bars for each subfigure.

[Figure]

**Figure S11.** Observed  $N_{ice \geq 100\mu m}$ (**top row**),  **(middle row)**, and $Q_{liq}$ (**bottom row**) for the sea ice (**column 1**), MIZ (**column 2**) and ocean (**column 3**) cases. Observations are shown as grey boxes. These boxes illustrate data similarly to those in Fig. 7. Modelled  $N_{ice>100\mu m}$, and $Q_{liq}$ are overlaid from the C86 (magenta), D10 (green), ACC (blue), D10×10 (red), and D10×0.1 (black) simulations. Model time steps of 21 h, 17 h, and 7 h are again used for comparison with the sea ice, MIZ, and ocean observations respectively.

[Figure]

**Figure S12.** Residual comparison of modelled and observed $N_{\overline{\text{ice}}\,\text{isg}}$ (**top row**), $N_{\text{ice}>100\mu m}$ (**middle row**), and $Q_{\text{liq}}$ (**bottom row**) in case 1 (sea ice) for each model time step. At each altitude bin, the mean observed quantity is subtracted from the mean modelled. The absolute magnitude of this fraction is then subtracted from 1. Therefore, better agreement between the mean observed and mean modelled values gives a larger fraction (with a maximum of 1). When two of the three parameterisations give good agreement with the $N_{\overline{\text{ice}}\,\text{isg}}$ observations at the same time step, that time step has been selected for comparison with the observations in Fig. 7. For the sea ice simulations, the chosen time step was 21 h. Note changing colour bars for each row.

[Figure]

**Figure S13.** Residual comparison of modelled and observed $N_{ice\ isg}$ (**top row**), $N_{ice>100\mu m}$ (**middle row**), and $Q_{liq}$ (**bottom row**) in case 2 (MIZ) for each model time step. As with Fig. S12, better agreement with the mean observed value gives a larger fraction (with a maximum of 1). For the MIZ simulations, the chosen time step was 17 h. Note changing colour bars for each row.

[Figure]

**Figure S14.** Residual comparison of modelled and observed $N_{\overline{ice}\ isg}$ (**top row**), $N_{ice>100\mu m}$ (**middle row**), and $Q_{liq}$ (**bottom row**) in case 3 (ocean) for each model time step. As with Fig. S12, better agreement with the mean observed value gives a larger fraction (with a maximum of 1). For the MIZ simulations, the chosen time step was 7 h. Note changing colour bars for each row.

[Figure]

**Figure S15.** Summed snow and graupel number concentrations ($N_{s+g}$, 1) and rain number concentration ($N_{liq}$, 2) using D10 (**top row**), ACC (**middle row**) and D10×0.1 (**bottom row**).  **(a, d, g):** Sea ice (case 1),  **(b, e, h):** MIZ (case 2),  **(c, f, i):** Ocean (case 3). Run length 24 hours.  Number concentrations of solid precipitable particles increase with simulation time in all cases when using D10, and the rain number concentration behaves similarly in case 2 when applying D10×0.1. Overall,  small concentrations of large solid and liquid  hydrometeors are modelled during the ACC simulations (panels **d, e, f**), and almost no  precipitable particles are modelled in case 1 with D10×0.1 (panel **g**). Note changing colour bar for each column.

[Figure]

**Figure S16.** Simulated ice number concentrations ($N_{\text{isg}}$, 1) and liquid water mixing ratios ($Q_{\text{liq}}$, 2) using ACC without large-scale subsidence (**top row**) and with an imposed subsidence of $2.5 \times 10^{-6}\,\text{s}^{-1}$ (**bottom row**, as in Solomon et al., 2015). All are restricted to water-saturation.  (a, d): Sea ice (case 1),  (b, e): MIZ (case 2),  (c, f): Ocean (case 3). Run length 24 hours. In all cases, cloud top height and $Q_{\text{liq}}$ is suppressed when large-scale subsidence is imposed. Temperatures are also warmer; however, case 2 is still too cold with comparison to the observations. Note changing colour bars for each subfigure.

[Figure]

**Figure S17.** Number concentrations of solid (snow + graupel, $N_{s+g}$, 1) and liquid (rain, $N_{rain}$, 2)  precipitable particles modelled during the **(a)** D10, **(b)** D10×0.5, and **(c)** D10×0.1 simulations over the ocean (case 3). Note changing colour bars for each subfigure.

[Figure]

**Figure S18.** Modelled LWP (**top row**), IWP (**second row**), and W (at approximately 1500 m, **bottom row**) for domain sizes 128 × 128 grid points (**left column**) and 196 × 196 grid points (**right column**) at 21 h into the simulations. Both domains use X/Y resolution of 120 m and use the same vertical domain size and resolution; the only difference is the domain size in X/Y. Convective cells – as shown by the hot-spots in LWP, IWP, and W – form in both cases, suggesting that these phenomena were not a result of the original domain specifications. Note changing colour bars for each subfigure.

[Figure]

**Figure S19.** Modelled LWP (**left panel**) and IWP (**right panel**) with time for the original domain size (128 × 128 grid points, **green**) and the larger domain size (196 × 196 grid points, **black**) over the ocean (case 3). These traces diverge at approximately 18 h; however, similar trends are seen. The feedbacks associated with convection and precipitation formation affect the evolution of the cloud properties, leading to different LWP and IWPs. These differences are due to the influence of the domain size on, for example, cloud radiative cooling and entrainment, leading to the formation of different convective cells, of different sizes, to the original domain.

[Figure]

**Figure S20.** Modelled $N_{\overline{ice}\ isg}$ (1) and $Q_{liq}$ (2) when using **(a)** D10  and **(b)** D10×0.1  to simulate case 2 over an extended run time of 45 h. Note changing colour bars for each subfigure.

[Figure]

**Figure S21.** Modelled LWP (red) and IWP (black) when using D10 (**solid**) and D10×0.1 (**dashed**) to simulate case 2 over an extended run time of 45 h.

[Figure]

**Figure S22.** Modelled LWP (**first column**), IWP (**second column**), and vertical velocity (**third column**) at approximately 1500 m using D10 (**top row**) and D10×0.1 (**bottom row**) to simulate case 2. Planar X-Y slices are shown at 37 h. Note changing colour bars for each subfigure.

[Figure]

**Figure S23.** Modelled number concentrations of solid ($N_{s+g}$, 1) and liquid ($N_{rain}$, 2)  precipitable particles when using **(a)** D10  and **(b)** D10×0.1 (**bottom row**) to simulate case 2 over an extended run time of 45 h. Note changing colour bars for each subfigure.